



# Also tropical freshwater ostracods show a seasonal life cycle

Juliane Meyer[1], Claudia Wrozyna[1], Albrecht Leis[2], Werner E. Piller[1]

[1] University of Graz, Institute of Earth Sciences, NAWI Graz Geocenter, Heinrichstrasse 26, 8010 Graz, Austria
[2] JR-AquaConSol GmbH, Steyrergasse 21, 8010 Graz, Austria

5  *Correspondence to*: Juliane Meyer (juliane.meyer@uni-graz.at)

**Abstract.** Isotopic signatures of ostracod shells became common proxies for the reconstruction of paleo-environmental conditions. Their isotopic composition is the result of the composition of their host water and the phenology and ecology of the target species. The sum of spatial and temporal variations from environmental factors in the species habitat defines the maximum isotopic variation of a population during the time of their shell formation. Here we present isotopic signatures 10  ($\delta^{18}$O, $\delta^{13}$C) of living *Cytheridella ilosvayi* (Ostracoda) and chemical and isotopic compositions of 14 simultaneously sampled freshwater habitats in South Florida and instrumental data of the region. The chemical and isotopic compositions of the selected sites characterize the different habitats and show the influence of the source water, biological activity and the duration of exposure to the surface. Isotopic signatures of *C. ilosvayi* shells correlate well with the isotopic composition of their host waters. Within-sample variability of repeated isotopic measurements of ostracod shells reflect habitat dependent 15  ranges and indicate temperature and the $\delta^{18}$O composition of precipitation ($\delta^{18}$O$_{prec}$) as regional environmental factors responsible for the population variation. Instrumental data of water temperature and $\delta^{18}$O$_{prec}$ were used to calculate the monthly variation of a theoretical calcite in rivers of Florida showing distinct seasonal variations in values and ranges. Different configurations of the theoretical calcite were compared to the within-sample variability to identify possible calcification periods of *C. ilosvayi*. For a plausible calcification period the ostracod isotopic range has to correlate with mean 20  values of the theoretical calcite with a slight positive offset (vital effect) and the extension of the theoretical calcite range. The tested model suggests a seasonal calcification period of *C. ilosvayi* in early spring. The surprising seasonality of a tropical ostracod life cycle is probably coupled to the hydrologic cycle of Florida. The results of this study contribute to the application of ostracod isotopes in modern calibration studies and their potential use in paleontology.

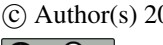



## 1 Introduction

Ostracod shells are discrete archives for biogenic carbonate and their chemical composition, including stable oxygen and carbon isotopes, are used routinely as proxies for environmental reconstructions (Von Grafenstein et al., 2000; Alvarez Zarikian et al., 2005; Anadón et al., 2006; Medley et al., 2008; Wrozyna et al., 2010; Escobar et al., 2012; Marco-Barba et

al., 2012; Pérez et al., 2013; Börner et al., 2013). Their shell is formed within a short time (hours to days) and reflect the conditions of the surrounding water ($\delta^{18}O_{water}$, $\delta^{13}C_{DIC}$, temperature) at the time of their formation with species specific vital effects (Xia et al., 1997a; Von Grafenstein et al., 1999; Keatings et al., 2002; Decrouy et al., 2011). The calibration of isotopic signatures of ostracods from modern water systems is necessary to understand the complex interaction of processes for a proper interpretation of the paleontological record.

In modern aquatic environments the stable oxygen and carbon isotopes are influenced by a variety of interlinked environmental factors that vary locally and seasonally (Schwalb, 2003; Leng and Marshall, 2004). These spatial and temporal changes in the environment result in an isotopic range of an ostracod population that can be characteristic for a specific site (Decrouy, 2012; Van der Meeren et al., 2011). Modern studies showed that changes in the within-sample variability of ostracods can be coupled to temporal (Heaton et al., 1995; Xia et al., 1997a; Meyer et al., 2016) and local (Von

Grafenstein et al., 1999; Meyer et al., 2016; Marco-Barba et al., 2012) differences of their environment.

The temporal environmental variation of a habitat reflected by isotopic range of ostracods is restricted to the duration of the calcification period of the target species. When seasonal shifts in the isotopic composition are indicative for certain time frames they can reveal insights on the annual timing and duration of the calcification period of an ostracod species. Thus, this information about the life cycle of an ostracod species can enhance the paleontological interpretation.

Here we present hydrological data and isotopic measurements ($\delta^{18}O$, $\delta^{13}C$) of *C. ilosvayi* from 14 modern freshwater habitats in South Florida. The aim of this study was to calculate potential calcification periods for the ostracod species *Cytheridella ilosvayi* Daday based on its $\delta^{18}O$ within-sample variability. Therefore, habitats were characterized by their chemical and stable isotope composition and compared to the isotopic signatures of *C. ilosvayi*. In addition, the isotopic within-sample variability of *C. ilosvayi* was compared between sites to identify parameters influencing their site specific isotopic range.

Further, instrumental hydrologic data of South Florida were used to set different seasonal scenarios for a theoretical calcite formed in equilibrium with the surrounding water. These settings were used to approximate potential calcification periods for *C. ilosvayi* based on the isotopic within-sample variability.



## 2 Study area

### 2.1 Sample locations

Numerous sinkholes, ponds, lakes, rivers, wetlands and marshes are widely spread over the whole Floridian peninsula. For this study 15 water and surface sediment samples were taken at 14 locations from different surface water bodies within the southern and southwestern watersheds in November 2013 and July/August 2014 (Fig. 1, Tab. 1).

In the southwestern watershed of Florida five samples were taken: four within the Peace River basin and a further one at Little Salt Spring in the Myakka River basin.

Saddle Creek is one of the head streams of Peace River that originates close to the City of Lakeland and then passes through Lake Hancock where the first sample was taken (LH) at a stillwater area close to the inflow of the lake. Onward, Saddle Creek empties into Peace River where it flows southwards and passes on to Charlotte Harbor estuary at Punta Gorda. South of the City of Bartow, in the center of Polk County, at one of numerous boat ramps of Peace River a second sample was taken in a stillwater area (PR). The southernmost tributary stream of Peace River is Shell Creek, which originates in the northwest of Charlotte County and flows westward where it converges with the Peace River at Punta Gorda. Sampling was performed two times (November 2013 and August 2014) in the littoral zone of an artificial dead end branch of the river at Hathaway Park (SC-3, SC-15).

Close to the City of North Port in the lower part of the Myakka River watershed Little Salt Spring (LSS) is located. It is a sinkhole of about 72 m depth discharging warm (~28 °C) slightly saline groundwater (Sacks and Tihansky, 1996). A sample was taken in the littoral zone of LSS.

The ten remaining locations are distributed within the southern watershed of Florida. Boundaries of the southern watershed are poorly defined. Lake Okeechobee is a central part of the hydrological region and receives water from precipitation and different surrounding river watersheds mainly from Kissimmee River in the north (Abtew, 2001).

The lake itself is surrounded by a complex system of drainage canals (Smith et al., 1989; Harvey and McCormick, 2009). This system is alternately used for controlled irrigation or drainage depending on the weather and season. Agriculturally used water can enter the lake at numerous locations of the canal system and increase the concentrations of nutrients of the lake (Gain, 1997). Sample CAL-1 and CAL-2 were taken from the littoral zone of drainage canals in the south of Lake Okeechobee. In the southwest of the lake, two canals converge at Moor Haven and flow into Caloosahatchee River where a further sample was taken in the littoral zone (CAL-4). Approximately 25 km downstream sample CAL-5 was taken from the littoral zone close to a boat ramp.

Much of the water leaving Lake Okeechobee to the south (surface and subsurface) forms huge areas of wetlands and marshes including the Everglades extending to the south of Florida (Meyers et al., 1993). At Rock Reef Pass Trail, a trail in the Everglades National Park located between the Shark River Slough and Taylor Slough in Dade County, periphyton was taken from the marsh (EG). Throughout the whole central Everglades conservation areas have been constructed using a levee and canal system for flood control and water supply which impedes the natural overland flow (Smith et al., 1989; Meyers et al.,



1993; Harvey and McCormick, 2009). Water allowances to the park are made through spillways along the Tamiami Canal in the north of the Everglades National Park. The Tamiami Trail Highway is one of the biggest impediments crossing the Everglades and adjacent areas from east to west. There, a further sample was taken at a marsh close to the trail in the Big Cypress National Preserve in Collier County (BiC).

Loxahatchee River is located in the east of the southern watershed and empties into the Atlantic Ocean at the City of Jupiter in Palm Beach County. There, four samples (LX-1, LX-2, LX-3 and LX-5) were taken in the littoral zone of Northwest Fork in short intervals within 2 km distance. In the sampling area Loxahatchee Rive receives its water from local shallow groundwater input, direct precipitation and runoff in the catchment (Swarzenski et al., 2006).

## 2.2 Hydrology

The Florida peninsula is a narrow carbonate platform (< 30 m elevation above sea level) mainly made up of thick sequences of limestones, dolomites and minor evaporites (gypsum and anhydrite) of Cretaceous to early Miocene age and underlain by volcanic, metamorphic and Mesozoic sedimentary rocks (Smith, 1982). The whole area is pervaded by a complex groundwater system that can be divided into three hydrogeological units: the surficial, the intermediate and the Floridian aquifer system (Maddox et al., 1992). The surficial aquifer system (SAS) is a shallow (less than 150 m) unconfined unit

bounded by the land surface. It is a freshwater aquifer and is locally used for agricultural and domestic water supplies, public water supply or municipal and industrial supplies (Maddox et al., 1992). The SAS is recharged primarily by seasonally fluctuating precipitation, but also by seepage from canals, other surface water bodies and the upward leakage of the intermediate aquifer system (IAS) (Wolansky, 1983; Alvarez Zarikian et al., 2005; Price and Swart, 2006). Local karst processes form open sinkholes or collapse features which are filled with SAS sediments which are in direct hydrologic

contact with the Florida aquifer system (Maddox et al., 1992). Saltwater intrusions into the SAS are possible along the coasts depending on the porosity and the permeability of the aquifer building a gradual mixing zone (Meyers et al., 1993; Wicks and Herman, 1995). From north to south the carbonate content of the aquifer increases and the calcium-carbonate water type dominates while in the coastal mixing zone sodium-chloride waters are dominant (Maddox et al., 1992). The base of the SAS is made up by an impermeable confining layer to the IAS. Water of the IAS is slightly saline with a chemical composition

varying from bicarbonate-dominated waters or mixed-ion type in the inland to sodium and chloride-dominated waters in coastal areas (Sacks and Tihansky, 1996). Its composition is controlled by the downward leakage from the SAS, the upward leakage from the Floridian aquifer system (FAS), aquifer mineral reactions and saltwater mixing (Sacks and Tihansky, 1996). The deepest unit is the FAS that can be separated locally into the upper and lower Floridian aquifer. Water of the FAS is generally brackish to saline in coastal areas. Its chemical composition ranges from calcium- and bicarbonate-dominated in

central recharge areas, to calcium-, magnesium- and sulfate-rich waters down gradient, and to a sodium-chloride type near the coast (Sacks and Tihansky, 1996). This aquifer unit is recharged by the IAS inland while in the coastal area upward discharge into the IAS is driven by the proximity to the sea (Sacks and Tihansky, 1996). Water from the IAS and the



Floridian aquifer system (FAS) is locally pumped to the surface for public water supply and agricultural usage (Wolansky, 1983; Alvarez Zarikian et al., 2005).Ut rutrum, sapien et vulputate molestie, augue velit consectetur lectus, bibendum porta justo odio lobortis ligula. In in urna nec arcu iaculis accumsan nec et quam. Integer ut orci mollis, varius justo vitae, pellentesque leo. Vestibulum eu finibus nisl. Cras ac arcu urna. Duis ut pellentesque urna.

## 5    2.3 Climate

The South Florida subtropical climate is characterized by highest air temperatures in August (22.6 – 33.0 °C) and lowest in January (10.4 – 23.0 °C) in the fifty-year average (Fig. 2). Single-digit temperatures during winter are possible, but frost is rare (Price et al., 2008). About 60-70 % of the rainfall in Florida occurs during the summer wet season from May to November in the course of thunderstorms and hurricanes with precipitation higher than 95 mm (Black, 1993).

The surface water level varies seasonally with the amount of precipitation. This results in increasing river runoff from June reaching its maximum in September. This also results in the drying-up of great areas of South Florida, including the Everglades, at the end of the wet season during spring (Duever et al., 1994).

The variation in evaporation follows a seasonal pattern, with lowest monthly evaporation occurring from December to February, and highest evaporation occurring from May to August (German, 2000). In the Everglades 70-90 % of water gets

evaporated with highest evaporation rates during summer while in spring evaporation is limited due to the low availability of water (Duever et al., 1994; Price et al., 2008). Evaporation of surface water contributes 7 – 12 % of the local atmosphere vapor (Price and Swart, 2006).

The major moisture source for precipitation in South Florida is evaporated seawater from the southeast of Florida. Sea-born vapor arises from the tropical North Atlantic dominating south Florida weather during the wet season by diurnally forced

sea-breeze, occasional easterly waves that originate from Africa and tropical cyclones that may drop tens of centimeters of precipitation in a single event as they make landfall (Price et al., 2008). Within the winter dry season Maritime-Tropical Air alternate with modified Continental-Polar Air from a high latitude North-American source (Price et al., 2008). When cold fronts from the north pass the peninsula sometimes intense precipitation occurs. Additionally, during the cool season westerlies can bring moisture from the Gulf of Mexico, Caribbean, or the tropical Western Pacific (Price et al., 2008).

Daily relative humidity in South Florida is about 70-80 % (Abtew, 2001; Price and Swart, 2006). The minimum humidity occurs midday and generally exceeds 50 % while maximum values occur by dawn and often approaches 90 % (Black, 1993; Abtew, 2001).

### 2.4 Seasonal variations of water isotopes

Several studies used the oxygen and deuterium isotopic composition of surface water and groundwater to assess the origin

and movement of water in Florida (Meyers et al., 1993; Sacks and Tihansky, 1996; Sacks, 2002; Wilcox et al., 2004; Price and Swart, 2006; Harvey and McCormick, 2009). These studies showed that precipitation is the major water source for



surface and groundwater in Florida and their isotopic composition is mainly affected by the seasonal isotopic variation of precipitation.

Seasonal δD and $\delta^{18}$O values of precipitation show high values during winter and low values at the start and end of the summer wet-season and slightly higher values in June through August (Price et al., 2008). The low isotopic values in the beginning of the summer suggest a more oceanic vapor source, fractionation by upstream rainout and greater disequilibrium between larger hydrometeors as they fall through lower-tropospheric isotope ratios (Price et al., 2008). In September and October tropical cyclones transport oceanic vapor to Florida and values decrease during that time again (Lawrence et al., 2004). Low isotopic values can also be observed in winter when cold fronts from middle-latitude North America pass Florida (Price et al., 2008). During the wet season surface water level increases and recycling of evaporated Everglades water increases the $\delta^{18}$O$_{prec}$ from June through October (Price and Swart, 2006).

The annual variability of $\delta^{18}$O values of waters in Florida is highest in surface waters and decreases with depth in the groundwater system (Meyers et al., 1993; Price and Swart, 2006). Evaporation of surface waters results in the accumulation of heavy isotopes along a local evaporation line (Swart et al., 1989; Meyers et al., 1993; Sacks, 2002; Harvey and McCormick, 2009). The $\delta^{18}$O values of shallow groundwater tend to vary with those of surface waters and net precipitation (rainfall minus evaporation) (Sacks, 2002; Price and Swart, 2006). When evaporated surface water enters the groundwater system it shows enriched in $^{18}$O values (Wilcox et al., 2004).

Data on the $\delta^{13}$C composition of inorganic carbon of water in Florida is rare. There are only data available from deeper groundwater that vary widely all over Florida from -14.9 to 0.54 ‰ (Sprinkle, 1989; Sacks and Tihansky, 1996). In general, $\delta^{13}$C values from the IAS are lighter compared to the FAS. Heavier $\delta^{13}$C values indicate dissolution of dolomite, which is more common in the deeper aquifer, while calcite precipitation removes some of the isotopically heavy isotopes (Sacks and Tihansky, 1996).

## 3 Material and Methods

### 3.1 Sediment samples and water analyses

Sampling and analysis methods are similar to Meyer et al. (2016). Surface sediment samples were taken to receive ostracod material. Simultaneously to water sampling, field variables (electrical conductivity, water temperature and pH) were measured *in situ* at all sample sites. Water samples were promptly filtrated using a syringe filter (pore size of 0.45 µm) and stored until analysis. Major ions, the isotopic composition of the water ($\delta^{18}$O, δD) and dissolved inorganic carbon ($\delta^{13}$C$_{DIC}$) were measured at the laboratory center of JR-AquaConSoL in Graz. The analytical procedure that was used in this study is similar to the method described by (Brand et al., 2009). A classic $CO_2$–$H_2O$ equilibrium technique (Epstein and Mayeda, 1953) with a fully automated device adapted from (Horita et al., 1989) coupled to a Finnigan DELTA$^{plus}$ Dual Inlet Mass Spectrometer for the measurement of oxygen isotopes. The stable isotopes of hydrogen in water were measured using a Finnigan DELTA$^{plus}$ XP mass spectrometer working in continuous flow mode by the chromium reduction method (Morrison



et al., 2001). Isotopic composition of DIC was analyzed using a Gasbench II device (Thermo) connected to a Finnigan DELTA$^{plus}$ XP isotope ratio mass spectrometer comparable to setups in other studies (Spötl, 2005). Results of isotopic measurements are given in per mil (‰) with respect to Vienna Mean Ocean Water (V-SMOW) and Vienna Peedee Belemnite (V-PDB), respectively, using the standard delta notation. The analytical precision for stable isotope measurements

is ±0.8 ‰ for δD, ±0.08 ‰ for δ$^{18}$O in water and ±0.1 ‰ δ$^{13}$C in DIC.

**3.2 Isotopic analyses of *Cytheridella ilosvayi***

Ostracods were picked from the sediment samples under a binocular (Zeiss Discovery V8) and shells of *Cytheridella ilosvayi* (Daday, 1905) were separated and stored in micro slides for isotopic measurements. *C. ilosvayi* was identified by morphological features of the shell in accordance with the description of (Purper, 1974).

Stable isotopic measurements were performed at the Institute of Earth Sciences, University of Graz. Per sample 1-16 measurements were performed for carbon and oxygen stable isotopes containing two to eight valves of *C. ilosvayi* (female, male and A-1) depending on the valve size and if fragments were missing. Adult and juvenile shells were analyzed separately. Prior to isotopic analyses soft part tissues and contaminations were removed from all ostracod valves with deionized water, brushes and entomological needles. If necessary, single valves were cleaned with H$_2$O$_2$ (10%) for five to ten

minutes at room temperature.

The samples were reacted with 100% phosphoric acid at 70 °C in a Kiel II automated reaction system and measured with a Finnigan DELTA$^{plus}$ isotope-ratio mass spectrometer. Reproducibility of replicate analyses for standards (in-house and NBS 19) was better than ±0.08 ‰ for δ$^{13}$C and ±0.1 ‰ for δ$^{18}$O. All carbonate isotopic values are quoted relative to V-PDB.

Isotopic values of *C. ilosvayi* were compared with their host water. Within-sample variability was evaluated for samples with

more than four measurements. This included LX-1, LX-2, LX-3, LX-5, CAL-1, CAL-2, CAL-4, CAL-5, PR, SC-3, SC-15 and EG.

**3.3 Calculation of the isotopic composition of calcite grown in equilibrium**

The isotopic composition of water and its temperature dependent fractionation (Kim and O'Neil, 1997) can be used to calculate the isotopic composition of a theoretical calcite precipitated in equilibrium as shown in the following equation:

$$\delta^{18}O_{calcite} = 1000 - \left( e^{\left(\frac{18.03*\left(\frac{1000}{T}\right)-32.42}{1000}\right)} * (1000 + \delta^{18}O_{water}) \right) \qquad (1)$$

where T is the measured water temperature in Kelvin.

Generally, δ$^{18}$O$_{water}$ values are expressed relative to V-SMOW, whereas δ$^{18}$O$_{calcite}$ values are expressed relative to V-PDB. To convert the δ$^{18}$O$_{water}$ values in the equation relative to V-PDB , the expression of (Coplen et al., 1983) was used.



### 3.4 Calculation of calcification periods

River and canal samples (LX-1, LX-2, LX-3, LX-5, CAL-1, CAL-2, CAL-4, CAL-5, SC-3, SC-15) were selected to compare the within-sample variability of *C. ilosvayi* with the calculated monthly range of a theoretical calcite grown under equilibrium conditions. To estimate plausible annual calcification times of *C. ilosvayi* a calcification period of one month
was assumed.

The calculation of the monthly range of an equilibrium calcite of a certain site was performed as following:

(a) Monthly $\delta^{18}O_{prec}$ ranges and monthly ranges of water temperature data were used in equation (1) to calculate the variation of a theoretical equilibrium calcite for one year.

(b) Measured $\delta^{18}O_{water}$ values from the investigated site and the mean monthly temperature from the sampling month of
the corresponding river were used to calculate the mean isotopic value of an equilibrium calcite precipitated in the particular aquatic system.

(c) The difference between the mean $\delta^{18}O_{prec}$ value from the sampling month and the river $\delta^{18}O$ value are calculated and all monthly $\delta^{18}O_{prec}$ values (min, max, mean) are corrected by that offset individually for each river (Fig. 3).

Daily temperature data were obtained from the National Water Information System Mapper (NWIS) of the US Geological
Survey (http://maps.waterdata.usgs.gov/mapper/) for Loxahatchee River (Station: 265906080093500) and Shell Creek (Station: 02297635) during the sampling period 2013/14 and for Caloosahatchee River (Station: 02292900) from May 2014 to April 2016. Data for Caloosahatchee River before 2014 are not available. Monthly minimum, maximum and mean temperature values are calculated from the included data years. Canal samples from Lake Okeechobee were related to Caloosahatchee River as no other data were available.

The $\delta^{18}O_{prec}$ composition was obtained from the Global Network for Isotopes in Precipitation (GNIP). Isotopic data ($\delta D$ and $\delta^{18}O$) of precipitation are available between 1997 and 2006 from 5 GNIP stations in South Florida. Spatially, inland sites are influenced by evaporated Everglades surface water while at the lower Keys precipitation is influenced by a more maritime vapor source (Price and Swart, 2006; Price et al., 2008).

For this study data from Rosenstiel School of Marine and Atmospheric Sciences (RSMAS), Biscayne National Park (BNP)
and Redlands GNIP stations were summarized and used to calculate monthly $\delta^{18}O_{prec}$ values between October 1997 and December 2006. We excluded the two coastal sites to minimize spatial influences assuming that recycling of Florida surface water influences precipitation in the whole study area equally.

The location of the GNIP and NWIS sites are displayed in Figure 1.





## 4 Results

### 4.1 Physico-chemical and stable isotope characteristics of the study sites

The results of all parameters measured in the field (temperature, pH, electrical conductivity (EC)), total dissolved solids (TDS) and salinity, as well as analysed in the laboratory (ions, $\delta D$, $\delta^{18}O_{water}$, $\delta^{13}C_{DIC}$) are summarized in Table 2 and Figures 4 to 6.

All investigated sites contain freshwater with salinity lower than 0.6 psu except for Little Salt Spring (LSS) which has a salinity of 2.6 psu (Tab. 2). The TDS of Loxahatchee River are similar between sampling locations (~ 275 mg/l) while samples of Lake Okeechobee Canal and Caloosahatchee River have higher values and cover a wider range (310.1-503.3 mg/l). The TDS value of BiC (464.0 mg/l) is most similar to CAL samples. Within the Peace River basin TDS values are lower for LH and PR (187.0 and 125.8 mg/l) than for Shell Creek in winter (655.8 mg/l) and in summer (490.2 mg/l). The TDS value of EG is most similar to PR.

Samples can be separated by their major anion and cation composition into three groups, calcium-bicarbonate-dominated, sodium-chloride-dominated and mixed waters (Fig. 4). Waters of the calcium-bicarbonate-type include EG, BiC, LH, CAL-5 and all Loxahatchee River samples. Samples belonging to the sodium-chloride-type are LSS and SC-15. The remaining samples (CAL-1, CAL-2, CAL-4, PR, SC-3) lie in a zone of mixing between these types.

Measured pH values range from 6.1 to 8.6. The majority of samples provide values between 7.0 and 7.9 except PR and LX-3 with lower values of 6.5 and 6.1, and EG and CAL-1 with higher values of 8.1 and 8.6, respectively.

All observed temperature ranges are in agreement with literature data for each site and season. Temperature measurements during winter provided values around 20 °C. One exception is Little Salt Spring showing raised winter temperatures of 26.8 °C. Temperatures range during summer from 28.3 to 35.5 °C. Values of Loxahatchee River show a variation of 1.3 °C between the locations during the sampling day. Samples from Lake Okeechobee Canal and Caloosahatchee River were also taken within one day and have a higher range of 5 °C.

The isotopic values of the water samples range from -6.0 to 16.9 ‰ for $\delta D$, from -1.74 to 2.35 ‰ for $\delta^{18}O$ and from -12.36 to -2.28 ‰ for $\delta^{13}C$ (Fig. 5 and 6, Tab. 2). The $\delta D$ and $\delta^{18}O$ values deviate negatively to the global meteoric water line (GMWL) and describe a local evaporation line (LEL) similar to the one described by Meyers et al. (1993) ($\delta D$=4,67($\pm$0,52) $\delta^{18}O$ + 2,68($\pm$3,86). The samples can be divided into three groups by their $\delta D$ and $\delta^{18}O$ values (Fig. 5). The first group includes the samples SC-15/SC-3 and LSS-1 with the lowest isotopic values (-6.0 to -4.12 ‰ for $\delta D$ and -1.74 to -1.28 ‰ for $\delta^{18}O$) that lie closest to the GMWL. The second group contains all other river samples and EG-3 and BiC-1 with values ranging from -2.4 to 4.9 ‰ and -0.73 to 0.40 ‰ for $\delta D$ and $\delta^{18}O$, respectively. LH, CAL-1 and 2 form the third group with the highest isotopic values (9.5 to 16.9 ‰ for $\delta D$ and 1.71 to 2.35 for $\delta^{18}O$). Concerning $\delta^{13}C_{DIC}$ the values can differ strongly between sites (Fig. 6a). Within the Loxahatchee River values are very similar (-10.60 to -9.70 ‰) while in Caloosahatchee River/Lake Okeechobee (-8.98 to -5.52 ‰) and the Peace River Basin (-12.36 to -6.62 ‰) the variation is strong. LSS has by far the highest value with -2.28 ‰.



## 4.2 Seasonal variation of theoretical equilibrium calcite

The increase of $\delta^{18}O_{prec}$ results in the same increase in the calculated calcite, while an increase in temperature results in a decrease in the $\delta^{18}O_{calcite}$ (Fig. 3). Thus, highest isotopic values for $\delta^{18}O_{calcite}$ can be calculated from minimum temperatures ($T_{min}$) and maximum $\delta^{18}O_{prec}$ ($O_{max}$) while the combination of $T_{max}$ and $O_{min}$ results in the lowest values (Fig. 3c).

Temperatures and $\delta^{18}O_{prec}$ vary differently throughout the year which leads to differences in the monthly values and ranges of equilibrium calcites.

Measured water temperatures of the investigated site lie all in the temperature range of the according NWIS stations except for SC-15 where the temperature exceeds the maximum value about 1 °C. However, the determined temperature ranges of the NWIS stations are suitable to approximate the range of an equilibrium calcite for the studied sites.

The annual variation of water temperatures of Shell Creek (SC), Caloosahatchee River (CAL) and Loxahatchee River (LX) at NWIS stations is correlated to annual air temperatures of Florida with highest values during the summer wet season and lowest during the winter dry season. But, the annual river temperature range is smaller than for air temperatures. Temperatures vary from 17.0 °C to 31.0 °C for LX, from 15.5 °C to 30.3 °C for SC, and from 17.0 °C to 37.0 °C for CAL (Tab. 3). Hence, $\delta^{18}O_{calcite}$ is high during winter and decreases until August where temperatures reach their maximum values

in all rivers.

The annual temperature variation ($T_{max}$-$T_{min}$) is similar for LX (14 °C) and SC (14.8 °C) and about 6 °C greater for CAL (20 °C). Temperatures in winter are similar for CAL and LX while SH has lower temperatures. In summer SC and LX are more similar while CAL has higher temperatures. This results in lowest $\delta^{18}O_{calcite}$ for CAL during summer and highest $\delta^{18}O_{calcite}$ for SC during winter.

The monthly temperature range for CAL is higher (5.0 to 8.6 °C) than for LX (4.4 to 7.6 °C) or SC (3.0 to 6.4 °C). A temperature increase of 1 °C can be translated into a decrease of 0.2 ‰ in the theoretical calcite (Craig, 1965; Chivas et al., 1986). Thus, the monthly isotopic range for a theoretical calcite caused by temperature differences within the rivers varies from 0.88 to 1.52 ‰ for LX, from 0.6 to 1.28 ‰ for SC and from 0.82 to 1.72 for CAL.

A distinct annual pattern in the monthly temperature ranges was not found. For instance, the range of Shell Creek is highest

in October and lowest in May while the highest range of Caloosahatchee River was found in August and the lowest in December (Tab. 3).

The annual mean value for $\delta^{18}O_{prec}$ is -1.94 ‰ and the total annual range varies from -10.31 ‰ in October to 1.53 ‰ in December (Fig. 3a; Tab. 4). Mean monthly values range from -3.43 ‰ in May to -0.75 ‰ in January. During the whole year values are relatively constant (-0.75 to -1.73 ‰). Only in the beginning (May/June) and the end (October/November) of the

wet season values fall below the annual mean value (-3.43 to -2.8 ‰). Maximum monthly $\delta^{18}O_{prec}$ values vary less then mean values, but, show also slightly lower values in the beginning and end of the wet season. The strongest variation can be seen in minimum values with a similar development throughout the year as mean and maximum values. The monthly $\delta^{18}O_{prec}$ range is correlated to minimum $\delta^{18}O_{prec}$ values. Negative excursions of $\delta^{18}O_{prec}$ cause the greater isotopic range. The lowest



variation has been observed at the end of the wet season during April (2.99 ‰) while the highest variation appears in October (10.18 ‰). The strong monthly variation of $\delta^{18}O_{prec}$ can also be seen in $\delta^{18}O_{calcite}$.

The increase of $\delta^{18}O_{prec}$ and the decrease of temperatures during winter both increase $\delta^{18}O_{calcite}$ values, while from July through September the temperature increase is contrary to the $\delta^{18}O_{prec}$ increase. This effect can be seen clearly in the

maximum $\delta^{18}O_{calcite}$ where the variation of precipitation is small and changes in temperature are more dominant. The variation of minimum $\delta^{18}O_{calcite}$ values is dominated by the variation of $\delta^{18}O_{prec}$. This results in highest values of $\delta^{18}O_{calcite}$ from December to March and lowest in the beginning and the end of the wet season in May/June and October. The lowest range was observed in the same month as for precipitation during April before the beginning of the wet season (3.97 ‰ for SC, 4.23 ‰ for CAL and 4.38 ‰ for LX) and the highest range in October (11.44 ‰ for SC, 11.25 ‰ for CAL and 11.32 ‰

for LX).

The enrichment of heavy isotopes of the investigated sites is considered to be constant during the year and seasonal changes of evaporation are neglected for the correction of their values. Hence, corrected $\delta^{18}O_{calcite}$ values of the study sites have the same seasonal variation as the theoretical equilibrium calcite and differ only in their annual offsets to each other.

### 4.3 Isotopic signatures of ostracod calcite

Isotopic values of *C. ilosvayi* ($\delta^{18}O_{ostr}$, $\delta^{13}C_{ostr}$) range from -3.05 to 2.28 ‰ for $\delta^{18}O_{ostr}$ and from -10.31 to -2.71 ‰ for $\delta^{13}C_{ostr}$, respectively (Tab. 5; Fig. 6b to 6f). There is a positive correlation between the mean $\delta^{18}O_{ostr}$ and their host waters (R²=0.66) and the mean $\delta^{13}C_{ostr}$ and the dissolved inorganic carbon (R²=0.90) (Fig. 7). This correlation for $\delta^{18}O_{ostr}$ becomes more significant excluding the strongly anthropogenic influenced CAL-4 from the statistical analyses (R²=0.83).

The highest mean $\delta^{18}O_{ostr}$ values were found in canal samples of Lake Okeechobee (0.05 to 1.25 ‰). River and marsh

samples from the southern watersheds show mean values in a similar range (-0.22 to -1.30 ‰). Within the Peace River watershed values decrease from north to south. Mean $\delta^{18}O_{ostr}$ values of SC-15 and LSS are the lowest with -2.09 ‰.

All river samples show equally low mean $\delta^{13}C_{ostr}$ values (-9.10 to-7.95 ‰) while canal samples and Lake Hancock exhibit higher values (-7.03 to -6.08 ‰). Both marsh samples have distinct different $\delta^{13}C$ values, whereat BiC is similar to river samples (-9.04 ‰ $\delta^{13}C$) and EG is more similar to canals and rivers (-6.01 ‰ $\delta^{13}C$). LSS has by far the highest $\delta^{13}C_{ostr}$ value

(-2.61 ‰).

Within-sample variability (max-min) of rivers and canals is relatively high for $\delta^{18}O_{ostr}$ and lower for $\delta^{13}C_{ostr}$. Canal samples CAL-1 and CAL-2 show the lowest $\delta^{18}O_{ostr}$ variation (1.04 ‰ and 1.21 ‰), while CAL-4 (2.44 ‰) has a variation similar to Loxahatchee River, Shell Creek and Caloosahatchee River (1.97 to 3.00 ‰). PR is the only river sample with a distinct lower $\delta^{18}O_{ostr}$ range (0.36 ‰) than all other samples. The $\delta^{18}O_{ostr}$ variation of the marsh sample EG is slightly higher

(3.08 ‰) than for the highest river variation.

The pattern for $\delta^{13}C_{ostr}$ is similar with lowest values for CAL-1 (0.47 ‰) and CAL-2 (0.65 ‰), while CAL-4 (1.62 ‰) is more similar to most river samples. The $\delta^{13}C_{ostr}$ variation of river samples range from 1.41 to 1.67 ‰ for Loxahatchee River, CAL-5 and Shell Creek in summer, while PR is more similar to canal samples (0.76 ‰). The range of the Shell Creek winter





sample is twice as high as the summer sample (2.95 ‰). EG has by far the highest $\delta^{13}C_{ostr}$ variation (5.16 ‰). Although there are just two measurements for BiC available, the $\delta^{13}C_{ostr}$ variation from the marsh sample is already higher (2.55 ‰) than for most river and canal samples.

# 5 Discussion

## 5.1 Physico-chemical and isotopic characteristics of surface water habitats

The major water source for any surface or subsurface water in South Florida is precipitation (Price and Swart, 2006). Precipitation has a generally low $\delta D$ and $\delta^{18}O$ composition and a low ion concentration with a composition similar to seawater of a sodium-chloride water type (Price and Swart, 2006).

The ion composition of precipitation is altered when it gets in contact with the underground. Dissolution of carbonates shifts the ionic composition of shallow groundwater from a sodium-chloride-dominated water type to a bicarbonate-dominated composition. Florida surface waters results from mixing between these two extreme water types (Price and Swart, 2006; Harvey and McCormick, 2009). Water from the SAS can seep into the lower groundwater aquifers. In the IAS and FAS also intrusions of seawater are possible and the ionic composition can change again to a sodium-chloride composition while isotopic composition again remains low.

The sodium-chloride-dominated water type, a salinity of ~ 2.6 psu, an elevated sulfate concentration (>250 mg/l), a nearly neutral pH value (7.5) (Fig. 4, Tab. 2) and the depth of ~70 m suggest that LSS is in contact with the lower part of the IAS (Alvarez Zarikian et al., 2005). The ion concentrations of deeper groundwater are higher than in LSS, but mixing with shallow groundwater and surface water runoff entering the sinkhole can lead to the observed values of LSS. In addition, groundwater in this region has annual temperatures of 24 to 28°C (Maddox et al., 1992), which correspond with measured temperatures of LSS (26.8 °C) in winter while temperatures from other surface water during that time had values about 6 °C lower (Tab. 2).

The only sample with a similar chemical composition is SC-15 which suggests a similar groundwater source. In contrast, SC-3 has a different chemical composition similar to PR. Within the Peace River Basin groundwater flows from the central part in the north to the coastal area in the south-west (Sacks and Tihansky, 1996). Lake Hancock in the north reflects the bicarbonate-dominated composition of the SAS in central Florida. Its low TDS value of <200 mg/l and the neutral pH (7.0) are typical for water from the SAS where carbonate dissolution already takes place but water gets periodically diluted by precipitation. Groundwater entering the rivers mixes with water from surface water runoff in the watershed shifting the ion proportion to a mixed water type which is reflected by PR and SC-3. The shift in the chemical composition of Shell Creek between summer and winter displays seasonal differences in the groundwater source of the river. In this region near the coast a change in the potentiometric surface of the FAS can cause a reversal between recharge and discharge of FAS and IAS and saltwater intrusions from the south appear periodically (Sacks and Tihansky, 1996).




Samples, that are located in the center of the southern watershed (CAL-1, CAL-2, CAL-4) receive water from Lake Okeechobee. The chemical composition of CAL-1 corresponds very well with water from Lake Okeechobee with an electrical conductivity of 400-500 µS/cm, a mixed water type (Harvey and McCormick, 2009). The higher conductivity of CAL-2 and CAL-4 implies additional water input from agricultural areas (median values of 946 µS/cm) surrounding the lake

(Harvey and McCormick, 2009). In addition, CAL-4 has raised phosphate (0.2 mg/l) and nitrate (0.48 mg/l) values compared to other canal samples that indicate agricultural usage.

Canal water gets discharged into Caloosahatchee River, but the change in the chemical composition to a bicarbonate-dominated water type of CAL-5 portrays the change in the watershed and increasing inflow of shallow groundwater to the river. The Everglades also receive water from the direction of Lake Okeechobee as surface and groundwater flow and from

local precipitation. The long water pathways from Lake Okeechobee to the Florida Keys cause big spatial and seasonal differences in the water composition (Meyers et al., 1993; Price and Swart, 2006; Harvey and McCormick, 2009).

The bicarbonate-dominated water type and the low TDS of EG differs from samples influenced by Lake Okeechobee water. Thus, local precipitation has a higher influence on the coastal area than water from Lake Okeechobee. BiC shows a distinct higher TDS concentration than EG and an ionic composition shifted to a more mixed water type. Big Cypress Swamp has a

different watershed then the Everglades what may explain the difference between BiC and EG. Both watersheds show similar hydrological conditions, and long water pathways and the local input of precipitation probably also causes spatial difference within Big Cypress Swamp. The bicarbonate-dominated water type, the low TDS, a neutral pH and isotopic values of Loxahatchee River are typical for a river receiving water from the SAS recharged by local precipitation and without the influence of inflowing water from Lake Okeechobee. The similarity of LX samples show a regional equal water

input from precipitation and groundwater along the river, that can also be seen in the isotopic composition.

The $\delta^{18}O$ composition of surface waters in Florida evolves through time as a response of evaporation. The longer the exposure of water to the surface, the higher is the accumulation of $^{18}O_{water}$ (Fig. 5). Samples that are dominated by groundwater inflow have the lowest $\delta^{18}O_{water}$ values and include LSS and SC-3 and SC-15. These samples have higher isotopic values than water from the FAS, SAS or precipitation indicating also a low influence by evaporated water from the

watershed. The change in the groundwater source between SC-3 and SC-15 is not reflected in the $\delta^{18}O_{water}$ values of this site. High $\delta^{18}O_{water}$ values are characteristic for lakes like Lake Hancock. CAL-1 and CAL-2 are strongly influenced by water from Lake Okeechobee showing also high $\delta^{18}O_{water}$ values. The chemical composition of CAL-4 corresponds with CAL-1 and CAL-2 but the sample has a much lower $\delta^{18}O_{water}$ value. Great differences can be seen in the isotopic composition between several canals around Lake Okeechobee. Canals with high isotopic values range in the dimension of the lake while

other canals have lower isotopic values and reflect additional components of runoff by flushing of rainfall and input of groundwater through agricultural area (Harvey and McCormick, 2009). This may explain the difference in the investigated canal samples and would also be consistent with higher conductivity values of CAL-4 (Harvey and McCormick, 2009). CAL-5 has a much lower $\delta^{18}O_{water}$ value (-0.73 ‰) indicating the addition of groundwater and the low influence of Lake Okeechobee water on the river. The remaining river and marsh samples lie between values of CAL-4 and CAL-5 and range





from -1 to 1 ‰ for $\delta^{18}O_{water}$. The through flow of rivers replaces water permanently and the accumulation of heavy isotopes is low. Varying retention times and periodic input of precipitation at BiC and EG can lead to big spatial and seasonal differences in the isotopic composition.

The $\delta^{13}C_{DIC}$ composition of the investigated sites differs widely (-12.36 to -2.28 ‰, Fig. 6a). In freshwater habitats the $^{13}C$ content depends on its source of dissolved $CO_2$ in the water from carbonate rock weathering, mineral springs, the atmosphere or respired organic matter (Peterson and Fry, 1987). Inflowing shallow groundwater and river waters have typically low values of -15 to -10 ‰ for $\delta^{13}C_{DIC}$ deriving from plant respiration and production of $CO_2$ in the soil. Loxahatchee River, CAL-5, PR, SC-15 and SC-3 reflect low values (-12.36 ‰ to -8.88 ‰) of incoming shallow groundwater. Discrepancies between the rivers are probably a result of the relative strength of $CO_2$ production in soil within the catchment areas or the exchange of $CO_2$ with the atmosphere (Atekwana and Krishnamurthy, 1998). Higher values in groundwater (-3 ‰ to +3 ‰) occur when limestone dissolution from the catchment is more dominant (Leng and Marshall, 2004). LSS is the only sample reflecting dissolution of older marine limestone from the deep groundwater aquifer with the highest observed $\delta^{13}C_{DIC}$ value (-2.28 ‰). The change to a deeper groundwater source in Shell Creek between summer and winter may also explain the shift to a more positive $\delta^{13}C_{DIC}$ value at SC-3. Long exposure of surface water leads to the exchange of $CO_2$ with the atmosphere until equilibrium at ~2.5 ‰ for $\delta^{13}C_{DIC}$ (e.g., Leng and Marshall 2004). None of the investigated samples has reached equilibrium with the atmosphere, but samples of Lake Hancock and CAL-1 and CAL-2 that are strongly connected to Lake Okeechobee, are more enriched in $\delta^{13}C_{DIC}$ indicating the longer residence time compared to rivers. BiC and EG are both characterized by a low water level, stagnant water and a dense aquatic vegetation. In such habitats the $\delta^{13}C_{DIC}$ composition depends strongly on the proportion of photosynthesis and respiration of aquatic organisms. In general, during photosynthesis $^{12}C$ concentration is reduced in the water by the preferential uptake of organisms resulting in high $\delta^{13}C_{DIC}$ values, while respiration has the opposite effect (e.g., Leng and Marshall 2004). The lower value of BiC may be the result of a lower photosynthetic activity of aquatic plants during winter. Further, EG was marked by a dense cover of periphyton. Algal fractionation during carbon uptake is higher than for aquatic plants (Rounick and Winterbourn, 1986) resulting in higher $\delta^{13}C_{DIC}$ values of the water and may explain the higher value of EG.

## 5.2 Stable isotopes of *Cytheridella ilosvayi*

Simultaneous sampling of ostracods and water can be used to relate them to each other on a regional scale when the number of samples and the isotopic range of host waters are high enough (Wetterich et al., 2008). For this study, the $\delta^{18}O$ and $\delta^{13}C$ composition of *C. ilosvayi* from 15 surface waters in Florida is strongly correlated to the $\delta^{13}C_{DIC}$ ($R^2$=0.90) and to a minor degree to the $\delta^{18}O$ of their host water ($R^2$=0.66). Thus, the number of sites and the isotopic range of surface waters were sufficient to connect mean isotopic values of *C. ilosvayi* with simultaneously taken water samples. The correlation of *C. ilosvayi* to $\delta^{18}O_{water}$ gets even more significant ($R^2$=0.83) excluding sample CAL-4 from the statistics (Fig. 7). As discussed above, CAL-4 is the only sample directly influenced by the temporal anthropogenic input of agricultural water which results in a distinct different isotopic composition compared to other canal samples with the same chemical composition. Contrary,



the isotopic composition of *C. ilosvayi* from CAL-4 is very similar to that of CAL-1 (Tab. 5, Fig. 6d). Thus, the input of agricultural water was probably initiated in the period between valve calcification and sampling, resulting in a great difference between $\delta^{18}O_{water}$ and $\delta^{18}O_{ostr}$. Further information on the anthropogenic usage of the canal system and the calcification time of *C. ilosvayi* would be needed to verify that.

It can be also expected from all sites that the water conditions (temperature, $\delta^{18}O_{water}$, $\delta^{13}C_{DIC}$) changed between valve calcification and sampling time. In open systems water input and output can be complex and different water bodies may behave seasonally different depending on the hydrologic factor dominating the system (DeDeckker and Forester, 1988; Leng and Marshall, 2004). The good correlation of ostracod and water isotopes then can indicate a calcification time close to sampling, a low seasonal variation of the habitats, or a similar seasonal habitat independent development of the sites between

calcification and sampling. Thus, CAL-4 is probably the only sample where the natural variation of the habitat gets strongly altered by the anthropogenic water input. At the repeatedly sampled site of Shell Creek (SC-3, SC-15) a shift can be observed in the water $\delta^{13}C_{DIC}$ from -8.88 ‰ to -10.73 ‰ between summer and winter, while the $\delta^{13}C$ of *C. ilosvayi* remains almost the same between seasons. This may indicate a calcification period during a time of the year where the hydrological conditions in the underground were similar. Still, it is unclear at which time *C. ilosvayi* formed their valve and at which point

the hydrological conditions of Shell Creek changed.

To overcome the uncertainty of time lags more information about the life cycle including molting periods of *C. ilosvayi* are necessary. But, if samples provide enough ostracod material to perform repeated isotopic measurements for one or more ostracod species, the intra- and interspecific variations can be useful to identify major changes in the environment during the time of calcification.

## 5.2.1 Within-sample variability

Two major factors are important for the isotopic composition of a single ostracod shell: the ostracod biology, determining the time (calcification period) and place (micro-habitat) of calcification, and the general characteristics of the environment itself, responsible for the seasonal variations (Decrouy, 2012). Small-scale differences of these factors result in the isotopic variation of an ostracod population at a specific site during the period of their valve calcification (within-sample variability).

Thus, the isotopic variation of a population is controlled by (1) the duration of the calcification period (Decrouy, 2012), (2) the seasonal environmental variation of a waterbody (Xia et al., 1997a; Von Grafenstein et al., 1999; Decrouy, 2012), and (3) the response of the micro-habitat to certain environmental changes (Decrouy, 2012).

The within-sample variability of *C. ilosvayi* was investigated from eight river sites, three canals and one marsh sample. All investigated micro-habitats are characterized by shallow water areas and a dense macrophyte cover (Tab. 1). EG is the only

lentic water body and accumulations of heavy or light isotopes are more probable in the marsh with a longer residence time than in rivers and canals with a permanent through flow.

EG exhibits a similar $\delta^{18}O_{ostr}$ variation as all river samples (except PR) and CAL-4, with variations of 1.97 to 3.08 ‰ (Tab. 5; Fig. 6).  In contrast, canal samples CAL-1 and CAL-2 have a smaller $\delta^{18}O_{ostr}$ range with 1.04 ‰ and 1.21 ‰. The similar



variation of the $\delta^{18}O_{ostr}$ ranges in Loxahatchee River can only be explained by a seasonally homogeneous isotopic development of the host water within the catchment area of the river. Furthermore, a regional influence that is independent from the catchment area seems to be a reasonable explanation for the similar ranges of LX, other river samples and EG. The seasonal temperature variation is similar at the whole peninsular and can vary strongly within hours in aquatic habitats with

small water volumes or a low water level. This will lead to a high variation in the ostracod calcite within a short time (Leng and Marshall, 2004), but it cannot explain the similarity of EG and rivers while the canal samples show lower ranges. It is more likely that another regional important factor, like the source water, causes the difference between canals and other sites. Rivers and EG are mainly fed by precipitation or surficial groundwater. The water source for the SAS is also precipitation and exhibits a similar isotopic range (Price and Swart, 2006). The permanent replacement of water in rivers results in a direct

reaction to changes in the isotopic composition of precipitation. Contrary, canal samples receive their water mainly from Lake Okeechobee and their $\delta^{18}O_{water}$ composition corresponds with the $\delta^{18}O$ range of the lake water of about 1 ‰ (Harvey and McCormick, 2009). In Lake Okeechobee incoming rainwater gets mixed with a great volume of older evaporated water buffering the $\delta^{18}O_{water}$ variation and explains the low $\delta^{18}O_{ostr}$ variation in canals. This is also in accordance with the assumption that CAL-4 receives water not only from Lake Okeechobee but also from agricultural areas that obtain water

from precipitation. Lentic water bodies with a smaller volume and a low water level (like marshes) have a much smaller buffering capacity and react similarly strong as rivers to changes in precipitation. This indicates the regional influence of precipitation as source water for surface waters in Florida (Price and Swart, 2006). Interestingly, the winter and summer samples of Shell Creek (SC-3 and SC-15) have a similar $\delta^{18}O_{ostr}$ range of 2.52 ‰ and 2.04 ‰. Together with a similar mean value, this indicates similar water and temperature conditions during the valve calcification time in both years and, thus, hint

to a seasonal calcification time of *C. ilosvayi*.

*C. ilosvayi* exhibits clear differences in the $\delta^{13}C_{ostr}$ range (Tab. 5; Fig. 6) between samples with a through flow and marsh samples indicating complex interactions of biological characteristics, input from external sources and mixing. Depending on the dominant source of carbon the $\delta^{13}C_{DIC}$ can vary widely within different time scales. Photosynthetic activity will remove $^{12}C$ from the system, while respiration has the opposite effect. The proportion of respiration and photosynthesis varies

between day and night and affects aquatic systems strongly with a high biological activity. Strong biological activity can be expected from all investigated habitats. The population of EG shows a variation higher (5.16 ‰) than for canals and rivers (0.47 ‰ to 2.95 ‰). The hydrological conditions at BiC are similar to EG and although there are just two measurements for this sample available, the $\delta^{13}C_{ostr}$ variation from the marsh is already higher (2.55 ‰) than for most river and canal samples. High residence time in marshes enables the accumulation and consumption of organic matter in the system which is probably

reduced in rivers and canals by their permanent flow. In addition, exchange of $CO_2$ with the atmosphere will increase the $\delta^{13}C_{DIC}$ over time. This process is also more important in marshes than in flowing water systems.

In rivers and canals large-scale processes like the input and mixing of inorganic carbon from different sources in the catchment is more important than local small scale processes (Atekwana and Krishnamurthy, 1998). This results in the small $\delta^{13}C_{ostr}$ range of 1.14 to 1.67 ‰ in Loxahatchee River, CAL-5 and SC-15. But, changes in the influx, e.g. increased input of



water from tributary streams after a rain shower, can cause shifts in the $\delta^{13}C_{DIC}$ composition. At SC-3 the $\delta^{13}C_{ostr}$ is higher (2.95 ‰) than in other river samples what is probably connected to the seasonal change in the groundwater source in the watershed that is not reflected in the $\delta^{18}O_{water}$ variation (Sacks and Tihansky, 1996). Further, CAL-1 and CAL-2 have a lower $\delta^{13}C_{ostr}$ range than rivers (0.47 ‰ and 0.65 ‰). The low variation is probably related to the dominance of inflowing Lake

Okeechobee water with a more stable $\delta^{13}C_{DIC}$ composition than in rivers, which have multiple tributary streams. CAL-4 has an $\delta^{13}C_{ostr}$ similar to river samples (1.62 ‰). Mixing of Lake Okeechobee water with agricultural water probably increases the variation of $\delta^{13}C_{ostr}$. PR is the only sample with an $\delta^{13}C_{ostr}$ range as small as CAL-1 and CAL-2 (0.76 ‰). Possibly the variation of the source water is small at this site or a higher flow rate stabilizes the $\delta^{13}C$ variation. It would also be possible that the number of measurements is too low to reveal statistically significant information from that site (Holmes, 2008).

Information on the life-history of *C. ilosvayi* is almost non-existent. It is unclear if this species has preferential molting periods for different development stages or if the population structure remains the same over the year. (Pérez et al., 2011) stated that surface sediment samples collected in November 2005 from Lago Petén Itzá (Guatemala) contained mainly valves of *C. ilosvayi* without soft parts while samples retrieved in February and March 2008 had both carapaces with soft parts, mostly from females. In Shell Creek we found a similar population variation with living *C. ilosvayi* very abundant in

summer and less in winter. This indicates a seasonal calcification period of *C. ilosvayi*. It is possible that climatic differences can cause discrepancies in the population structure of a species from different sites (Schweitzer and Lohmann, 1990). But, within the region of South Florida climatic variation is negligible and calcification periods of *C. ilosvayi* should be equal at all sites. Then, it can be expected that the within-sample variability from a single species provide information of a similar time frame. Consequently, when the seasonal variation of a habitat is strong enough the within-sample variability of *C.*

*ilosvayi* contains information on the time and duration of its calcification period.

## 5.3 Reconstruction of *C. ilosvayi* calcification times from rivers using within-sample $\delta^{18}O$ variability

To calculate a plausible calcification time for *C. ilosvayi* during a year we used instrumental data of water temperatures and $\delta^{18}O_{prec}$ to determine possible monthly compositions of an equilibrium calcite precipitated in rivers and canals of Florida and compared it to the within-sample range of ostracods (Fig. 8).

Two requirements have to be complied for a plausible calcification time: (1) the $\delta^{18}O$ range of the equilibrium calcite has to correspond with the $\delta^{18}O_{ostr}$ range and (2) a positive vital effect has to be considered for the $\delta^{18}O_{ostr}$ (Xia et al., 1997b; Von Grafenstein et al., 1999; Decrouy et al., 2011).

The calculated monthly mean $\delta^{18}O_{calcite}$ values and ranges vary seasonally distinctly among each other and can be characteristic for certain months. In general, calcite ranges exceed the $\delta^{18}O$ range of *C. ilosvayi* in every sample and month

(Fig. 8). From the beginning of the wet season in May until December the range is up to three times higher than for *C. ilosvayi*. A shorter calcification period of half a month will not reduce the isotopic range of the theoretical calcite significantly. For instance, within October the temperature range of CAL in the first half of the month is 2.3 °C smaller than for the whole month. For $\delta^{18}O_{prec}$ the range will be reduced about 1 ‰. Thus, the $\delta^{18}O_{calcite}$ range will be reduced from



11.25 ‰ to 9.79 ‰ and remain much higher than the ostracod range. Hence, months with a high $\delta^{18}O_{calcite}$ range can be excluded as calcification period for *C. ilosvayi*. From January onward the range decreases constantly until it reaches its minimum in April. During April the $\delta^{18}O_{calcite}$ range varies between 3.97 ‰ in SC and 4.38 ‰ in LX and is most similar to the ostracod $\delta^{18}O$ ranges.

For a plausible calcification time it can be expected, that values of *C. ilosvayi* lie within the range of the theoretical calcite and tend to more positive values compared to mean calcite values due to a vital effect (Xia et al., 1997a; Xia et al., 1997b; Von Grafenstein et al., 1999; Decrouy et al., 2011). A vital effect for modern *C. ilosvayi* of +1.04 ‰ is reported from Lake Petén Itzá (Escobar et al., 2012). Currently, it has been shown, that the vital effect within a species can differ between sites and the ionic composition of the water may change vital effects (Marco-Barba et al., 2012; Decrouy and Vennemann, 2013).

However, the chemical composition of the investigated sites is considered to be stable enough for a constant vital effect within a sample. Investigations on possible seasonal changes of the vital effect from a single site lie beyond the methodical approach of this study.

During the wet season from May until October positive offset of *C. ilosvayi* to the mean calcite value exceeds 1.04 ‰ by far and even exceeds maximum values of the theoretical calcite at nearly all sites excluding these months as calcification time

(Fig. 8). In November, values of the theoretical calcite increase until April and the $\delta^{18}O_{ostr}$ values converge to mean calcite values with a lower positive offset.

The combination of a small range and a ~1 ‰ positive vital offset between *C. ilosvayi* and the theoretical calcite assuming a maximum one month calcification period indicates April as the most plausible calcification time for *C. ilosvayi* (Fig. 8). This would also fit with the finding of *C. ilosvayi* from Guatemala where it molts in spring (Pérez et al., 2011).

The applied model is restricted to the variation of extreme values of two components influencing the final composition of river waters. Other factors like mixing of different source waters, the variation of evaporation, or the anthropogenic regulation of a water body probably influence the actual isotopic range of aquatic habitats. Further, large rain amounts with low $\delta^{18}O_{prec}$ values from thunderstorms may have a stronger influence on the isotopic composition of surface water then small amounts of precipitation with high values. But, low $\delta^{18}O_{prec}$ values are not exclusively connected to heavy rainfalls and

can also occur during the winter dry season. Thus, the listed factors cannot be included in the calculation without further investigations. However, assuming a lower isotopic variation of precipitation during the sampling years would result in a lower variation of the theoretical calcite and other month become also plausible as calcification period. This would be the case for the period from January to March where the isotopic variation would decrease to a range comparable to *C. ilosvayi* and vital offsets are also reasonable. Nevertheless, calcification will remain seasonal and can be excluded to take place

during the summer wet season in Florida.

Various kinds of life cycles are known from freshwater ostracods (Cohen and Morin, 1990). These include seasonal cycles, multiple cycles with or without overlapping generations and non-seasonal continuous life cycles of ostracods. The isotopic range of *C. ilosvayi* indicates clearly a restricted seasonal calcification period. However, almost nothing is known about the controlling factors for life cycles of tropical freshwater ostracods. In temperate and boreal regions temperature is regarded as



the main abiotic factor controlling seasonal ostracod population dynamics (Horne, 1983; Cohen and Morin, 1990). In cooler climates it is necessary to overcome sub-zero temperatures during the cold season, but this is not the case in warm regions like Florida of Guatemala. Further, low temperatures slow down the development of ostracods and increase inter-molt periods which could be the case in Florida winter (Martens, 1985). But this would not explain a restriction of the calcification period to a certain season. Other factors like variation of food supply, water conditions or competition have also been suggested to influence the periodicity of ostracods (Horne, 1983; Kamiya, 1988; Martens, 1985). These factors are possibly coupled to other abiotic factors than temperature. In both regions, Florida and Guatemala, the wet season lasts from May to October. The initiation of the rainy season in spring leads to flooding of dried up areas and higher surface runoff which has an essential influence on seasonal habitat conditions and input of organic matter as food source. The connection to the hydrological cycle is a plausible explanation for the seasonality of the life cycle of a tropical ostracod. *C. ilosvayi* possibly overwinters the dry season in a juvenile stage and maturation is initiated during early spring when rain sets in, water level and food supply rise and, thus, conditions for reproduction are more advantageous.

In a paleontological application the within-horizon stable isotope variability of *C. ilosvayi* from lake sediment cores on the Yucatan Peninsula has been successfully used as proxy for high-frequency climate variability (Escobar et al., 2010). In that study a low within-sample $\delta^{18}O$ variability of ostracod shells could be connected with periods of constant aridity when the ratio of evaporation/precipitation was also low and supports the suggestion of a life cycle connected to the hydrological cycle.

# 6 Conclusions

In this study we present a model to calculate the calcification period of the ostracod species *Cytheridella ilosvayi* from South Florida. This model is based on the comparison of instrumental hydrological background data and simultaneously taken water samples and living ostracods.

The combination of the isotopic and chemical composition of surface waters shows distinct patterns that characterizes the different habitats. The variation of the chemical composition of surface waters differed from a bicarbonate-dominated to a sodium-chloride-dominated water type as a result of solution processes at the surface and in the underground and illustrates the origin of the source waters for the study sites. Contrary, the $\delta D$ and $\delta^{18}O$ composition differs as a function of evaporation. Sites with the longest exposure to the surface showed the most positive isotopic values. Further, the $\delta^{13}C$ composition differs widely caused by the combination of different processes such as biological processes, limestone dissolution and degassing.

Simultaneous sampling of water and ostracods was useful to correlate the $\delta^{18}O$ and $\delta^{13}C$ signature of *C. ilosvayi* to their host water and expose the general isotopic characteristics ($\delta^{13}C_{DIC}$ and $\delta^{18}O_{water}$) of their habitats. Different isotopic offsets between the sites can be addressed to changes in the water input and output of the complex hydrologic systems between valve calcification and sampling. The within-sample variability of *C. ilosvayi* ($\delta^{18}O$, $\delta^{13}C$) from all samples provide



information on the same time duration and, thus, identify changes in the different environments during their shell calcification period. Similar $\delta^{18}O_{ostr}$ ranges of the samples show the regional equal influence of precipitation on marshes and rivers. The variation is smaller in canals due to the influence of older evaporated source water. High $\delta^{13}C_{ostr}$ variability in marsh samples is caused by rapid differences in the biological activity. In contrast, rivers and canals are dominated by input

and mixing of inorganic carbon from different sources in the catchment leading to small $\delta^{13}C_{ostr}$ ranges.

Monthly maximum ranges of $\delta^{18}O$ from a theoretical calcite in equilibrium with the surrounding water were calculated from instrumental data of river water temperatures and $\delta^{18}O_{prec}$. The composition of the theoretical equilibrium calcite varied seasonally with high mean values in winter and low values in summer. Ranges were highest in the beginning and end of the wet season and lowest in April in the end of the dry season. These monthly ranges were compared to the isotopic range of *C.*

*ilosvayi* to identify possible calcification times during the year. Using this scenario, the most plausible calcification period for *C. ilosvayi* is in April when water temperatures are high enough and the $\delta^{18}O_{prec}$ range is lowest. A seasonal calcification period is surprising for a tropical ostracod. However, this seasonality is probably connected to strong seasonal changes of habitat conditions caused by an annual weathering cycle. This model contributes to the use of ostracod isotopes as indicator for the phenology of modern ostracods and the potential extension of interpretations in paleontological studies using ostracod

isotopes.

## Data availability

All relevant data are presented within the manuscript or in supplementary material

## Author contribution

J. Meyer, C. Wrozyna and W.E. Piller carried out sampling of all water and sediment material and measurements of field
data. In addition, J. Meyer and C. Wrozyna prepared ostracod material for isotopic analyses and J. Meyer carried them out. A. Leis carried out water analyses. J. Meyer prepared the manuscript with contributions from all co-authors.

## Competing interests

The authors declare that they have no conflict of interest.

## Acknowledgements

The study was financed by the Austrian Science Fund (FWF) project P26554-N29. We thank Martin Gross (Universalmuseum Joanneum) for his support during sampling 2013 and useful discussion on the topic of this study. Sylvain Richoz (University of Graz) is thanked for support on stable isotope measurements and useful discussion.





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





**Tables**

Table 1 Location and characterization of the studied sites

| Sample | Date | Latitude | Longitude | Location | Water body type | Habitat | Sample depth (m) |
|---|---|---|---|---|---|---|---|
| LSS | 26.11.2013 | 27° 4' 29.33" | 082° 14' 0.37" | Little Salt Spring | sinkhole | littoral zone | 1.0 |
| SC-3 | 28.11.2013 | 26° 58' 27.04" | 081° 53' 21.55" | Shell Creek, Hatheway Park | artificial river branch | littoral zone | 0.3 |
| BiC | 29.11.2013 | 25° 53' 29.53" | 081° 16' 14.52" | Big Cypris National Reserve | marsh | swamp | 0.2 |
| LX-1 | 31.07.2014 | 26° 56' 03.00" | 080° 10' 36.40" | Loxahatchee River | river | littoral zone | 1.0 |
| LX-2 | 31.07.2014 | 26° 56' 32.50" | 080° 10' 19.20" | Loxahatchee River | river | littoral zone | 0.2 |
| LX-3 | 31.07.2014 | 26°56'40.28" | 080° 10' 15.94" | Loxahatchee River | river | littoral zone | 0.2 |
| LX-5 | 31.07.2014 | 26° 56' 49.80" | 080° 10' 12.40" | Loxahatchee River | river | littoral zone | 0.2 |
| EG | 02.08.2014 | 25° 26' 2.00" | 080° 45' 12.30" | Rock Reef Pass Trail, Everglades National Park | marsh | swamp, periphyton | 0.3 |
| CAL-1 | 06.08.2014 | 26° 43' 37.90" | 080° 42' 10.50" | Lake Okeechobee Canal | artificial canal | littoral zone | 0.25 |
| CAL-2 | 06.08.2014 | 26° 45' 41.50" | 080° 55' 11.70" | Lake Okeechobee Canal | artificial canal | littoral zone | 1.2 |
| CAL-4 | 06.08.2014 | 26° 50' 09.80" | 081° 05' 14.40" | Lake Okeechobee Canal | artificial canal | littoral zone | 0.3 |
| CAL-5 | 06.08.2014 | 26° 47' 21.70" | 081° 18' 33.60" | Caloosahatchee River | river | stillwater area | 0.1-0.4 |
| LH | 07.08.2014 | 28° 0' 7.310" | 081° 51' 4.22" | Lake Hancock | lake | lake inflow, stillwater area | 0.05-0.2 |
| PR | 07.08.2014 | 27° 48' 46.20" | 081° 47' 36.90" | Peace River | river | stillwater area | 0.5 |
| SC-15 | 08.08.2014 | 26° 58' 26.99" | 081° 53' 21.81" | Shell Creek, Hatheway Park | artificial river branch | littoral zone | 0.3 |



5    Table 2 Physicochemical and stable isotope characteristics of the studied sites

| Sample | Field parameters | | | | | Cations | | | | Anions | | | | | Stable Isotopes | | |
|---|---|---|---|---|---|---|---|---|---|---|---|---|---|---|---|---|---|
| | Temp [°C] | pH | EC [µS/cm] | Sal [psu] | TDS [mg/l] | Na [mg/l] | K [mg/l] | Mg [mg/l] | Ca [mg/l] | Cl [mg/l] | NO₃ [mg/l] | SO₄ [mg/l] | HCO₃ [mg/l] | Br [mg/l] | δD [‰] | δ¹⁸O [‰] | δ¹³C [‰] |
| LSS | 26.8 | 7.5 | 5110 | 2.6 | 3193.4 | 766.6 | 22.3 | 135.8 | 167.5 | 1430 | <0.25 | 498.9 | 172.1 | <0.5 | -4.12 | -1.28 | -2.28 |
| SC-3 | 20.3 | 7.9 | 938 | 0.56 | 655.8 | 71.9 | 5.1 | 13.3 | 103.1 | 145.0 | <0.25 | 59.8 | 257.5 | 0.6 | -5.9 | -1.40 | -8.88 |
| BiC | 20.8 | 7.6 | 586 | 0.35 | 464.0 | 32.8 | 1.6 | 6.2 | 87.3 | 56.9 | <0.25 | 4.8 | 274.6 | <0.5 | -0.89 | -0.54 | -9.55 |
| LX-1 | 30.6 | 7.3 | 363 | 0.22 | 270.1 | 22.8 | 1.38 | 2.64 | 50.09 | 35.84 | 0.42 | 6.64 | 150.7 | 0.07 | 2.4 | 0.28 | -9.70 |
| LX-2 | 30.5 | 7.2 | 375 | 0.23 | 276.6 | 23.0 | 1.36 | 2.62 | 51.73 | 36.26 | 0.44 | 7.36 | 154.4 | 0.04 | 2.1 | 0.32 | -10.60 |
| LX-3 | 31.7 | 6.1* | 375 | 0.23 | 275.8 | 23.1 | 1.36 | 2.63 | 52.13 | 36.08 | 0.47 | 7.32 | 153.2 | 0.06 | 2.1 | 0.28 | -10.42 |
| LX-5 | 30.4 | 7.1 | 375 | 0.23 | 276.0 | 22.8 | 1.39 | 2.50 | 51.50 | 36.14 | <0.01 | 7.33 | 154.4 | 0.07 | 0.3 | 0.11 | -9.92 |
| EG | 33.1 | 8.1 | 189 | 0.11 | 152.8 | 5.2 | 0.25 | 0.65 | 33.30 | 7.20 | <0.01 | <0.1 | 106.2 | <0.01 | 5.8 | 0.19 | -6.13 |
| CAL-1 | 31 | 8.6 | 444 | 0.27 | 310.1 | 32.8 | 5.92 | 9.43 | 42.31 | 57.84 | <0.01 | 30.06 | 131.8 | 0.16 | 16.9 | 2.35 | -5.52 |
| CAL-2 | 30.5 | 7.4 | 676 | 0.4 | 466.1 | 51.7 | 9.46 | 13.14 | 65.81 | 85.72 | 0.17 | 47.46 | 192.8 | 0.21 | 13.0 | 1.72 | -7.81 |
| CAL-4 | 35.5 | 7.5 | 724 | 0.43 | 503.3 | 52.3 | 8.31 | 13.12 | 77.93 | 90.49 | 0.48 | 57.43 | 203.8 | 0.24 | 4.9 | 0.40 | -8.31 |
| CAL-5 | 34.7 | 7.4 | 550 | 0.33 | 394.4 | 25.8 | 6.21 | 7.00 | 74.78 | 45.35 | 0.23 | 30.77 | 204.4 | 0.14 | -1.3 | -0.73 | -8.98 |
| LH | 28.3 | 7.0 | 247 | 0.14 | 187.0 | 12.2 | 3.27 | 5.44 | 29.01 | 20.02 | 0.46 | 2.90 | 114.1 | 0.01 | 9.5 | 1.71 | -6.62 |
| PR | 28.3 | 6.5 | 189 | 0.11 | 125.8 | 14.4 | 5.36 | 4.30 | 14.27 | 20.64 | 1.45 | 14.96 | 51.9 | 0.05 | -2.4 | -0.28 | -12.36 |
| SC-15 | 31.2 | 7.1 | 297* | 0.18 | 490.2 | 105.3 | 5.35 | 14.31 | 40.99 | 193.8 | 0.11 | 34.74 | 95.8 | 0.59 | -6.0 | -1.74 | -10.73 |

*- accuracy of *in situ* measurement uncertain



Table 3 Monthly temperature data from NWIS stations: 02297635 (Shell Creek), 02292900 (Caloosahatchee River) and 265906080093500 (Loxahatchee River)

| | Shell Creek | | | | Caloosahatchee River | | | | Loxahatchee River | | | |
| --- | --- | --- | --- | --- | --- | --- | --- | --- | --- | --- | --- | --- |
| | Mean | Max | Min | Max-Min | Mean | Max | Min | Max-Min | Mean | Max | Min | Max-Min |
| Jan | 17.7 | 20.4 | 15.5 | 4.9 | 20.6 | 25.0 | 17.6 | 7.4 | 20.5 | 23.3 | 17.0 | 6.3 |
| Feb | 18.1 | 21.1 | 16.2 | 5.0 | 19.3 | 22.4 | 17.0 | 5.4 | 21.7 | 25.0 | 17.4 | 7.6 |
| Mar | 19.8 | 21.7 | 18.3 | 3.4 | 24.3 | 28.0 | 19.4 | 8.6 | 21.5 | 25.0 | 18.1 | 6.9 |
| Apr | 22.7 | 24.7 | 19.9 | 4.8 | 27.1 | 30.8 | 24.5 | 6.3 | 25.2 | 28.3 | 21.4 | 6.9 |
| May | 25.1 | 26.7 | 23.7 | 3.0 | 28.9 | 31.7 | 26.7 | 5.0 | 26.6 | 29.3 | 22.7 | 6.7 |
| Jun | 27.3 | 29.8 | 24.9 | 4.9 | 30.6 | 34.3 | 28.3 | 6.0 | 28.2 | 30.7 | 25.7 | 5.1 |
| Jul | 28.0 | 30.3 | 25.8 | 4.5 | 30.9 | 34.5 | 28.7 | 5.9 | 28.5 | 30.9 | 26.2 | 4.7 |
| Aug | 28.4 | 30.3 | 26.2 | 4.1 | 31.0 | 37.0 | 28.4 | 8.6 | 29.3 | 31.0 | 26.7 | 4.4 |
| Sep | 27.1 | 29.1 | 24.6 | 4.5 | 29.7 | 33.8 | 27.6 | 6.2 | 27.8 | 30.4 | 26.0 | 4.4 |
| Oct | 25.0 | 28.2 | 21.8 | 6.4 | 27.7 | 31.0 | 25.4 | 5.6 | 26.2 | 29.0 | 23.2 | 5.9 |
| Nov | 20.6 | 22.9 | 17.9 | 5.1 | 24.6 | 28.0 | 21.7 | 6.3 | 22.5 | 25.5 | 18.9 | 6.6 |
| Dec | 18.5 | 21.0 | 16.4 | 4.6 | 22.2 | 24.4 | 20.3 | 4.1 | 21.3 | 23.7 | 18.2 | 5.6 |





Table 4 Monthly values of $\delta^{18}O$ from precipitation of southeast Florida including data from GNIP stations: Biscayne National Park BNP, Rosensteil School of Marine and Atmospheric Sciences (RSMAS) and Redland 1998-2005

| | $\delta^{18}O$ | | | | | Number of measurements | | | |
|---|---|---|---|---|---|---|---|---|---|
| Month | Mean | Max | Min | Max-Min | SD | BNP | RSMAS | Redland | total |
| Jan | -0.75 | 0.62 | -3.81 | 4.43 | 1.45 | 6 | 5 | 1 | 12 |
| Feb | -1.40 | 1.11 | -3.36 | 4.47 | 1.18 | 2 | 10 | 2 | 14 |
| Mar | -1.17 | 1.47 | -4.08 | 5.55 | 1.44 | 2 | 11 | 3 | 16 |
| Apr | -1.14 | 0.75 | -2.24 | 2.99 | 1.22 | 0 | 7 | 0 | 7 |
| May | -3.43 | -0.49 | -7.62 | 7.13 | 1.97 | 1 | 6 | 5 | 12 |
| Jun | -2.80 | 0.12 | -8.72 | 8.84 | 2.11 | 5 | 8 | 5 | 18 |
| Jul | -1.69 | 0.9 | -4.46 | 5.36 | 1.58 | 5 | 17 | 2 | 24 |
| Aug | -1.56 | 0.86 | -4.22 | 5.08 | 1.42 | 4 | 18 | 4 | 26 |
| Sep | -2.81 | 0.94 | -5.57 | 6.51 | 1.70 | 4 | 9 | 3 | 16 |
| Oct | -3.37 | -0.13 | -10.31 | 10.18 | 2.70 | 12 | 13 | 5 | 30 |
| Nov | -1.73 | 0.61 | -6.03 | 6.64 | 1.53 | 15 | 10 | 1 | 26 |
| Dez | -1.40 | 1.53 | -7.43 | 8.96 | 1.88 | 28 | 15 | 5 | 48 |





Table 5 Isotopic data of *Cytheridella ilosvayi* in comparison to its host water

| Sample | Water | | | | *Cytheridella ilosvayi* | | | | | | | | | | |
| | | | | | $\delta^{18}O_{valve}$ (‰ V-PDB) | | | | | $\delta^{13}C_{valve}$ (‰ V-PDB) | | | | |
| | $\delta^{18}O$ (‰ V-SMOW) | $\delta^{13}C_{DIC}$ (‰ V-PDB) | Temp (°C) | $n^a$ | Mean | STD | Min | Max | Max-Min[b] | Mean | STD | Min | Max | Max-Min[b] |
|---|---|---|---|---|---|---|---|---|---|---|---|---|---|---|
| LSS | -1.28 | -2.28 | 26.8 | 1 | -2.09 | - | - | - | - | -2.61 | - | - | - | - |
| SC-3 | -1.40 | -8.88 | 20.3 | 8 | -1.80 | 0.96 | -3.05 | -0.53 | 2.52 | -8.17 | 0.98 | -9.59 | -6.64 | 2.95 |
| BiC | -0.54 | -9.55 | 20.8 | 2 | -1.27 | 0.02 | -1.29 | -1.26 | (0.03) | -9.04 | 1.80 | -10.31 | -7.76 | (2.55) |
| LX-1 | 0.28 | -9.70 | 30.6 | 8 | -0.22 | 0.77 | -1.78 | 0.58 | 2.36 | -7.95 | 0.65 | -8.71 | -7.04 | 1.67 |
| LX-2 | 0.32 | -10.60 | 30.5 | 9 | -1.26 | 0.67 | -2.24 | -0.27 | 1.97 | -8.51 | 0.46 | -9.24 | -7.77 | 1.47 |
| LX-3 | 0.28 | -10.42 | 31.7 | 8 | -0.76 | 0.67 | -1.54 | 0.55 | 2.09 | -8.44 | 0.53 | -9.47 | -7.96 | 1.51 |
| LX-5 | 0.11 | -9.92 | 30.4 | 7 | -1.11 | 1.16 | -2.87 | 0.13 | 3.00 | -8.21 | 0.61 | -8.75 | -7.29 | 1.46 |
| EG | 0.19 | -6.13 | 33.1 | 7 | -1.30 | 1.32 | -2.42 | 0.66 | 3.08 | -6.01 | 2.12 | -7.87 | -2.71 | 5.16 |
| CAL-1 | 2.35 | -5.52 | 31 | 4 | 1.25 | 0.49 | 0.78 | 1.82 | 1.04 | -6.27 | 0.22 | -6.53 | -6.06 | 0.47 |
| CAL-2 | 1.72 | -7.81 | 30.5 | 6 | 0.05 | 0.45 | -0.68 | 0.53 | 1.21 | -6.08 | 0.25 | -6.34 | -5.69 | 0.65 |
| CAL-4 | 0.40 | -8.31 | 35.5 | 16 | 1.12 | 0.76 | -0.16 | 2.28 | 2.44 | -7.03 | 0.51 | -7.86 | -6.24 | 1.62 |
| CAL-5 | -0.73 | -8.98 | 34.7 | 8 | -1.03 | 1.16 | -2.28 | 0.30 | 2.58 | -8.06 | 0.46 | -8.52 | -7.38 | 1.14 |
| LH | 1.71 | -6.62 | 28.3 | 1 | -0.29 | - | - | - | - | -6.36 | - | - | - | - |
| PR | -0.28 | -12.36 | 28.3 | 4 | -1.02 | 0.16 | -1.25 | -0.89 | 0.36 | -9.10 | 0.33 | -9.42 | -8.66 | 0.76 |
| SC-15 | -1.74 | -10.73 | 31.2 | 7 | -2.09 | 0.73 | -2.99 | -0.95 | 2.04 | -8.77 | 0.51 | -9.59 | -8.18 | 1.41 |

[a] n- number of measurements
[b] numbers in brackets are excluded from the discussion of the within-sample variability





**Figure 1: Location of sample sites (BiC=Big Cypress Swamp, CAL=Caloosahatchee River, EG=Everglades, LH= Lake Hancock, LSS=Little Salt Spring, LX= Loxahatchee River, PR= Peace River, SC=Shell Creek). Also included are GNIP stations (Redlands, RSMAS, BNP) and NWIS stations (NWIS 1=02297635; NWIS 2=02292900; NWIS 3=265906080093500).**



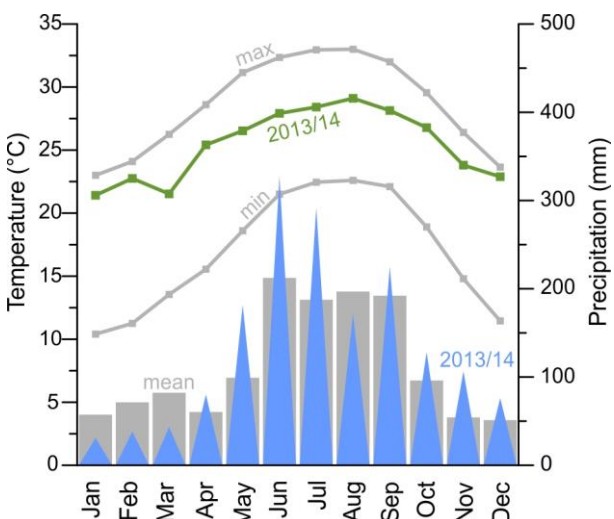

Figure 2: Fifty-years average (1955-2005) of maximum and minimum temperatures and mean precipitation of the southwestern and southern catchment area of Florida in comparison to mean temperature and mean precipitation of Miami 2013/14 (National Climate Change Viewer, U.S. Geological Survey; http://www2.usgs.gov/climate_landuse/clu_rd/nccv.asp)





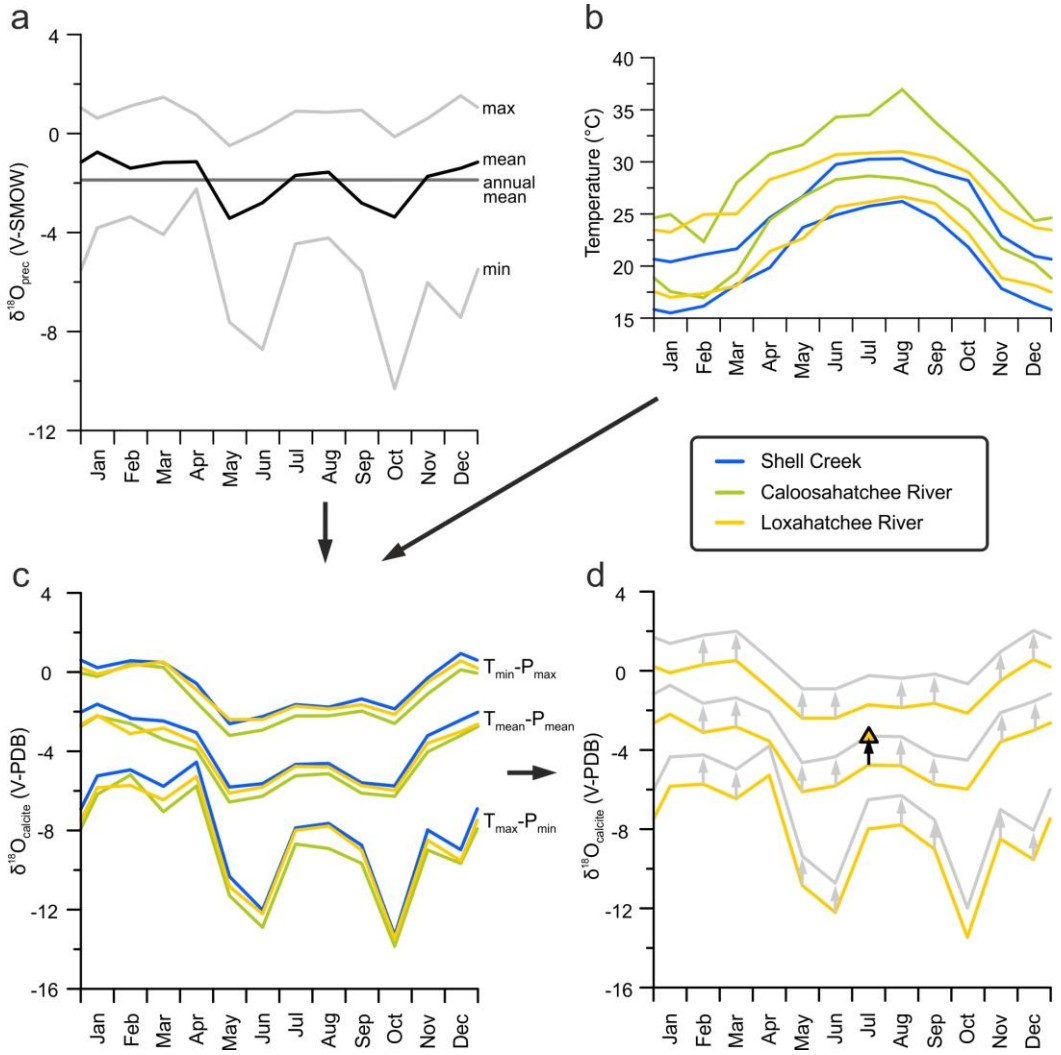

**Figure 3: Modeling of the monthly composition of a theoretical calcite formed in equilibrium in Florida rivers (a) precipitation δ18O; (b) water temperature from rivers in Florida; (c) calculated calcite ranges using equation (1); (d) example for the offset correction from LX-4. For detailed explanations see text (chapter 3.4).**





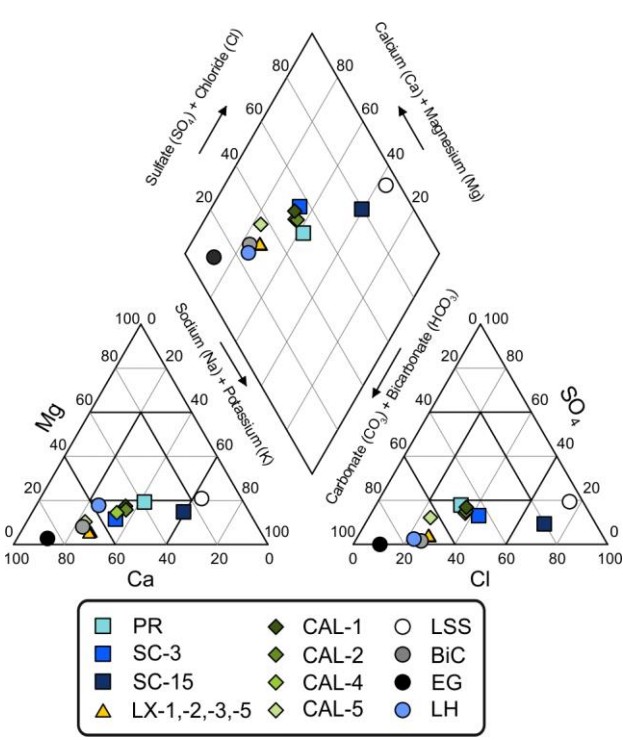

**Figure 4: Piper diagram illustrating the major ion composition of the investigated sites.**

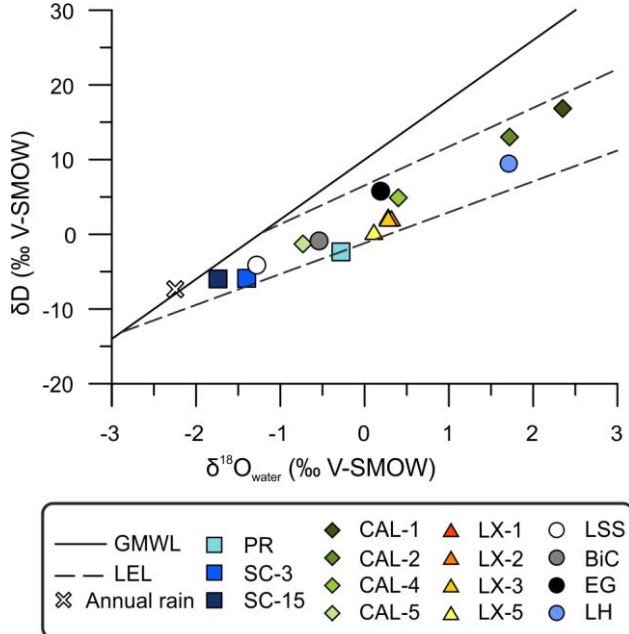

**Figure 5: Stable oxygen and deuterium isotope composition of all water samples in comparison to the Global Meteoric Water Line (GMWL) (Craig 1961) and Local Evaporation Lines (LEL) from Meyers et al. (1993). Also included is the annual rainfall calculated from GNIP stations.**





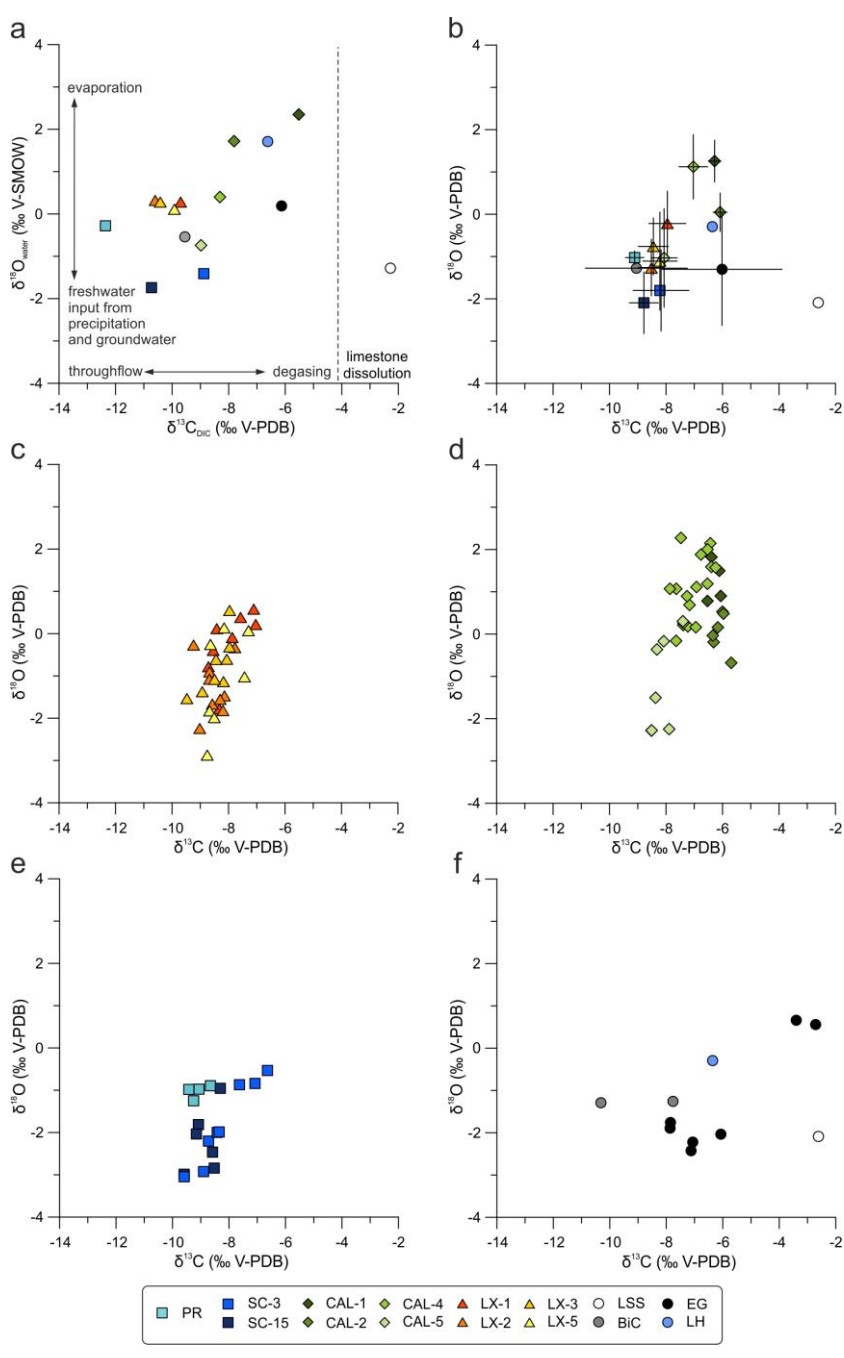

**Figure 6: δ13C and δ$^{18}$O values of water samples and *C. ilosvayi*. (a) water samples, controls on the isotopic composition of the sites are indicated; (b)-(f) *C. ilosvayi*: (b) mean values of all sites with standard deviation; (c) *C. ilosvayi* from Loxahatchee River; (d) *C. ilosvayi* from Lake Okeechobee Canal and Caloosahatchee River; (e) *C. ilosvayi* from Peace River and Shell Creek; (f) *C. ilosvayi* from Everglades, Big Cypress Swamp, Lake Hancock and Little Salt Spring**




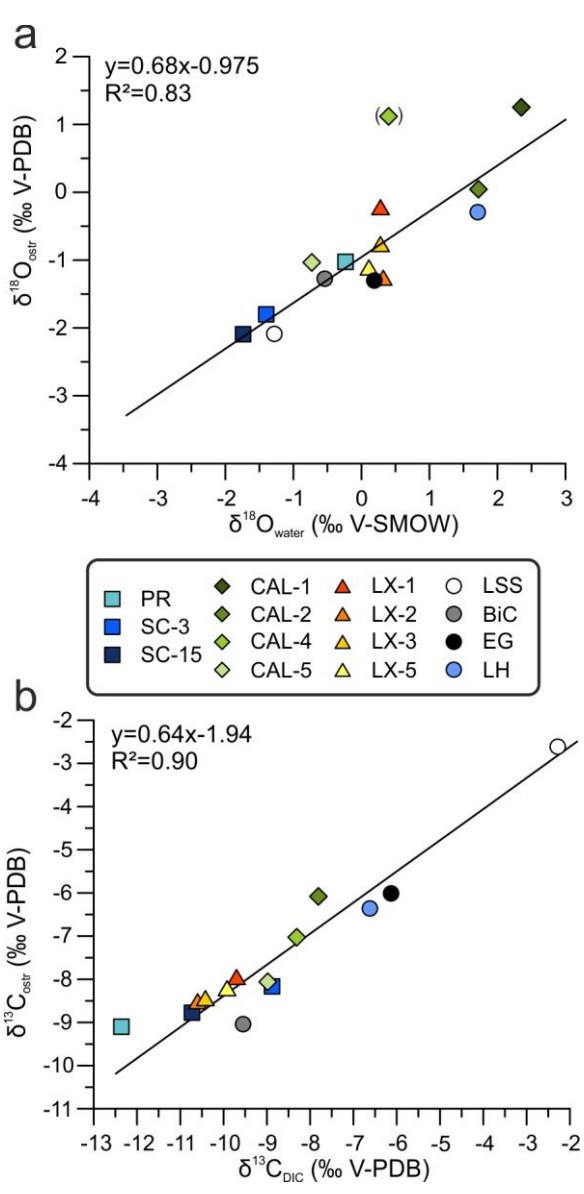

**Figure 7: Correlation of the isotopic composition of ostracod valves of *C. ilosvayi* vs. the water in which they evolved: (a) ostracod δ$^{18}$O vs. water δ$^{18}$O, sample in brackets is excluded from the correlation statistics, for further explanations see text; (b) ostracod δ$^{13}$C vs. δ$^{13}$C$_{DIC}$**







**Figure 8: δ¹⁸O of *C. ilosvayi* compared to δ¹⁸O range of a calcite in isotopic equilibrium calculated from water temperatures obtained from NWIS stations and precipitation δ¹⁸O from GNIP stations. Horizontal black lines indicate mean values of *C. ilosvayi* δ¹⁸O, black vertical lines indicate maximum and minimum values and black vertical bars show the standard deviation. Gray bars indicate the maximum range of δ¹⁸O of a calcite in isotopic equilibrium during the particular month.**