# Peer review of "Also tropical freshwater ostracods show a seasonal life cycle"

_Biogeosciences, 2017_

## Referee Comment (RC1) · J. Holmes (Referee) · 29 Mar 2017

**Overview**

Fossil non-marine ostracods are commonly used as sources of calcite for stable-isotope analyses. Although studies based on ostracod calcite are valuable, complications arising from vital offsets from isotopic equilibrium and uncertainties relating to the site and season of calcification can limit the value of such palaeo-isotope studies. Investigations into the modern isotope systematics of individual ostracod species, such as the one described here, are therefore of potential value and interest. This MS is concerned with the oxygen and carbon isotope analyses of specimens of the subtropical/tropical non-marine ostracod Cytheridella ilosvayi from surface-water sites in southern Florida, which also have been analysed for water chemistry and stable

isotope composition. Cytheridella ilosvayi is a fairly widespread Neotropical ostracod species, with potential for use in palaeo-isotope investigations. The MS presents an interesting dataset based on isotope analyses of modern collections of the species and its host water/DIC from sites in Southern Florida. These data are then used estimate the liklely calcification season of Cytheridella ilosvayi based on assumptions about seasonal changes in the inferred oxygen-isotope composition of calcium carbonate precipitated at isotopic equilibrium with the host water, and allowing for a positive offset from oxygen-isotope equilibrium based on a previous study. Despite the potential value of this work, I have a number of queries and concerns that I think need to be addressed before it can be considered for final publication.

Specific comments

Title. Southern Florida is subtropical, not tropical. The premise that the data can be used to argue that tropical ostracods show a seasonal lifecycle therefore needs to be re-evaluated and the title needs to be changed. The seasonal range of temperatures in Florida is quite large, for example, meaning that this might impact the timing of calcification of Cytheridella ilosvayi.

Abstract. The MS includes carbon isotopes, but these are hardly mentioned in the abstract.

Page 2 Line 22. Explain how carbon isotopes fit into the study.

Page 3-4 Section 2.1. You include much more detail about the sites than is needed for this paper. Some of the information could also be summarized in a table.

Page 5. Line 6. Here you refer to South Florida's climate as 'subtropical'

Page 7 Line 11. The 'bulked' samples, especially of 8 valves, will tend to cause averaging of the results and so reduce the variability. Previous studies investigating variability (e.g. Escobar et al., 2010 JOPL; Dixit et al., 2015, JOPL) have analyzed numerous single shells in order to assess variability. This point needs to be considered in the

discussion.

Page 10. Section 4.2. Modelling the equilibrium values in terms of the isotopic composition of rainfall and water temperature is unlikely to be realistic. On lines 11-13 on the next page, you state that enrichment of heavy isotopes is considered to be constant during the year. In the absence of data to support this statement, I do not think it can be regarded as valid. For example, Sachs (2002) (which you cite), states that 'The isotopic composition of Lake Starr and Halfmoon Lake varied seasonally (fig. 15), primarily in response to net precipitation (rainfall minus evaporation).' Admittedly these sites are from Central Florida, not Southern Florida: however, I still feel that you need to provide evidence to support your statement about seasonally-constant evaporation if you believe that this assumption is valid, or modify your discussion if you cannot support your statement.

Line 21. It is approximately 0.2 ‰ – the relationship is not linear. Also why cite Chivas et al. (1986) here? Why not cite the authors of the equation you use i.e. Kim and O'Neil (1997)?

Page 11 Line 26. Oxygen isotopes in your individual samples might be expected to show less variability than carbon isotopes because the latter are affected by small-scale microhabitat variations in the carbon-isotope composition of the DIC. The fact that you see the reverse could be argued to be linked to seasonal variations in evaporation, meaning that the actual water-isotope values vary more than you have modelled simply based on temperature and the isotopic composition of rainfall. In any case, comparing variability of carbon and oxygen is only really meaningful if you know the actual seasonal variation in water and DIC isotope values.

Page 12 Section 5.1. Much of the detail in this section can be cut as it is not really relevant to the main focus of the paper.

Page 13. Line 21. Doesn't this statement suggest that evaporation might also vary seasonally as well?

Page 14 Section 5.2 and figure 7. This is the key figure in the paper and it suggests to me that the specimens of Cytheridella ilosvayi may have calcified quite close to the sampling time – this would explain the fairly strong correlation between the measured isotope composition of the water and the DIC, and the isotopic composition of the ostracod shells.

Page 15-16 Line 33 (page 15) to line 2 (page 16). But surely this depends on the length of time over which the shells calcified. If they calcified over a short time in the season, then the degree of variability would be small even if site conditions changed seasonally. In addition, you need to consider the fact that you did not measure single shells (see my point above). This will have the effect of reducing the observed variability.

Page 16 Lines 11-12. I could not see evidence in the cited paper for a 1‰ range in lake water in Lake Okeechobee. For sure, there are a few datapoints from this lake in Figure 4 of Harvey and McCormick, but it is unclear whether these are representative of seasonal variations in the isotopic composition of Lake Okeechobee or are samples from different locations in the lake from one single sampling period. Also, by themselves, these data do not support the suggestion that the amount of evaporation of this lake is seasonally constant (see also my point above).

Page 17. Section 5.3. This method will only work if you can assume that the isotopic composition of water at each of your sites is closely coupled to the isotopic composition of rainfall. As noted above, you have very little evidence to support this. As I also note above, figure 7 is the key figure in the paper and it suggests to me that the specimens of Cytheridella ilosvayi may have calcified quite close to the sampling time. In this case, I wonder whether these theoretical calcite calculations are really very necessary.

Page 18. Lines 17-19. I am not convinced that this is valid, because your calculations of theoretic equilibrium calcite composition assume that you know the isotopic composition of the water at each site in detail, and I am not convinced that you do.

Line 34. Cytheridella ilosvayi may be a neotropical species, but Florida is a subtropical

locality and the seasonal variations in temperature, although certainly not as large as at higher latitudes, are not negligible. In any case, it is misleading to state that your work shows that 'tropical' ostracods show seasonality.

General comment. You emphasize oxygen-isotopes in your paper. Although the carbon-isotope results are referred to, your comments about these are more limited and speculative, not least because you have very little information about the variations in the carbon-isotope composition of the DIC. It would be useful to comment on the carbon-isotope data in a little more detail.

Summary evaluation

Although this is an interesting dataset and a potentially valuable contribution to the literature, the critical comments need to be addressed before it can be published.

Technical comments

Page 2 Line 3 and elsewhere. von Grafenstein (not Von) Line 10. No need to include the word 'stable' Page 5 Lines 2-4. What has happened here?

---

## Short Comment (SC1) · 26 Apr 2017

We thank the referee Jonathan Holmes for his constructive comments on the open discussion paper. Although, he thinks it is an interesting dataset there are some important points that needs to be explained and/or changed. Comments will be given point by point as received in the referee comments:

Title. Southern Florida is subtropical, not tropical. The premise that the data can be used to argue that tropical ostracods show a seasonal lifecycle therefore needs to be re-evaluated and the title needs to be changed. The seasonal range of temperatures in Florida is quite large, for example, meaning that this might impact the timing of calcification of Cytheridella ilosvayi.

> Florida is located on the boundary between the tropical and subtropical region. The

southern part of the state is defined as tropical, while the northern region is subtropical (Kottek et al., 2006). But, the transition between both climatic regions is gradual. Samples from southern watershed are all located in the tropical part of Florida, while samples from the southwestern watershed may belong to the subtropical region. However, a comparison of air temperatures and precipitation between the different watersheds did not show significant differences (we used the National Climate Change Viewer provided by the US Geological Survey) and were related to the same climate zone.

> The title will be changed from 'tropical' to '(sub-) tropical'.

> Of course our findings are limited to the northern edge of the tropical region, but the comparisons to a population of C. ilosvayi in Guatemala in the middle of the tropics also shows annual differences in their occurrence. > The influence of temperature is commented in section 5.3. But, additional discussion on the influence of temperature will be added.

Abstract. The MS includes carbon isotopes, but these are hardly mentioned in the abstract.

> Information on carbon isotopes will be added to the abstract.

Page 2 Line 22. Explain how carbon isotopes fit into the study.

> Aims of the study will be supplemented.

Page 3-4 Section 2.1. You include much more detail about the sites than is needed for this paper. Some of the information could also be summarized in a table.

> Section 2.1. will be abridged, relevant information on the sites will be integrated in table 1.

Page 5. Line 6. Here you refer to South Florida's climate as 'subtropical'

> A paragraph will be added to the section to point out the position of Florida at the

boundary between tropical and subtropical climate.

Page 7 Line 11. The 'bulked' samples, especially of 8 valves, will tend to cause averaging of the results and so reduce the variability. Previous studies investigating variability (e.g. Escobar et al., 2010 JOPL; Dixit et al., 2015, JOPL) have analyzed numerous single shells in order to assess variability. This point needs to be considered in the discussion.

> 'Bulking' is an important point comparing the variability of samples. We tried to avoid the use of multiple valves as far as possible. In most cases we used two adult valves for one measurement (if possible of the same individual) to avoid averaging. Especially in river samples, the number of single shell measurements is high. We tested the calculation of the optimal sample size (Holmes, 2008; Escobar et al., 2010; Dixit et al., 2015) for rivers (with a 10% acceptable error and a confidence level of 90%) and found, that the number of measurements from LX and CAL to be sufficient, while in Peace River the number of measurements may be too small.

> Respective comments will be given in the discussion on the averaging effect of 'bulked' samples and the number of measurements used for the comparison of within-sample variations.

> The number of used valves was stated wrong and will be corrected to 'one to nine' (page 7, line 11). Additionally, the number of valves used for each measurement will be added in the supplementary material.

Page 10. Section 4.2. Modelling the equilibrium values in terms of the isotopic composition of rainfall and water temperature is unlikely to be realistic. On lines 11-13 on the next page, you state that enrichment of heavy isotopes is considered to be constant during the year. In the absence of data to support this statement, I do not think it can be regarded as valid. For example, Sachs (2002) (which you cite), states that 'The isotopic composition of Lake Starr and Halfmoon Lake varied seasonally (fig. 15), primarily in response to net precipitation (rainfall minus evaporation).' Admittedly these sites are

from Central Florida, not Southern Florida: however, I still feel that you need to provide evidence to support your statement about seasonally-constant evaporation if you believe that this assumption is valid, or modify your discussion if you cannot support your statement.

> The model we used refers to rivers, not lakes. In lakes evaporation is much more important than in any streaming water. The isotopic composition of rivers is always described as a mixture of overland flow and baseflow deriving from groundwater (e.g., Criss 1999; Clark and Fritz, 1996). If a river is influenced by direct evaporation this would be seen in the gradual enrichment of heavy 18O isotopes along the river. But this is not the case in the rivers we investigated. However, the enrichment of heavy isotopes in river samples that can be seen in Fig. 5 of this manuscript probably results from the addition of groundwater that has been recharged by evaporated surface water (Meyers et al., 1993; Wilcox et al., 2004; Price and Swart, 2006). This may vary seasonally, but the exact processes for the seasonal recharge of groundwater are beyond the means of this study. However, Price and Swart (2006) showed that surface water from wells and canals and groundwater in the everglades varies seasonally and are event driven (Fig. 7) with highest values in the end of the dry season and lowest values during the wet season. This is contrary to the findings of Sacks et al. (2002) and indicates that precipitation is the major influence on the isotopic composition of small surface waters in summer. Evaporation of surface water may get more important in winter, when precipitation amounts are low. But, it also has to be noted, that $\delta$18O values of precipitation are also higher during the dry season.

> The respective sentence in page 10 will be deleted.

> Data on the isotopic composition of further river water samples will be added in a supplementary file.

> Comments on the possible influence of evaporation on the seasonal isotopic composition of the theoretical calcite will be given in the discussion.

Line 21. It is approximately 0.2 ‰ – the relationship is not linear. Also why cite Chivas et al. (1986) here? Why not cite the authors of the equation you use i.e. Kim and O'Neil (1997)?

> The reference will be changed.

Page 11 Line 26. Oxygen isotopes in your individual samples might be expected to show less variability than carbon isotopes because the latter are affected by small-scale microhabitat variations in the carbon-isotope composition of the DIC. The fact that you see the reverse could be argued to be linked to seasonal variations in evaporation, meaning that the actual water-isotope values vary more than you have modelled simply based on temperature and the isotopic composition of rainfall. In any case, comparing variability of carbon and oxygen is only really meaningful if you know the actual seasonal variation in water and DIC isotope values.

> The sentence will be deleted as we have no further information on the seasonal variation of DIC isotopes.

Page 12 Section 5.1. Much of the detail in this section can be cut as it is not really relevant to the main focus of the paper.

> Much of the details on the ionic composition were given in order to illustrate potential influences on the seasonal isotopic composition.

> Some details on the ionic composition of the sites will be summarized and shortened.

Page 13. Line 21. Doesn't this statement suggest that evaporation might also vary seasonally as well?

> As stated before, direct evaporation has a minor influence on rivers.

> Paragraphs will be added in the manuscript on the seasonal influence of evaporation on rivers.

Page 14 Section 5.2 and figure 7. This is the key figure in the paper and it suggests

to me that the specimens of Cytheridella ilosvayi may have calcified quite close to the sampling time – this would explain the fairly strong correlation between the measured isotope composition of the water and the DIC, and the isotopic composition of the ostracod shells.

> This is already considered in this section. If calcification took place close to the sampling, than the within-sample variability has to co-vary with the calculated theoretical calcite range before sampling. But, this is not the case. The further discussion on the within sample variability aims to examine if there is any period during the year where the ranges fit to each other.

Page 15-16 Line 33 (page 15) to line 2 (page 16). But surely this depends on the length of time over which the shells calcified. If they calcified over a short time in the season, then the degree of variability would be small even if site conditions changed seasonally. In addition, you need to consider the fact that you did not measure single shells (see my point above). This will have the effect of reducing the observed variability.

> I agree with the referee, the isotopic variation will depend on the length of time in which a species calcifies and will only display the variation of the environment during this time. The exact duration of this timescale is unclear, but considering the small climatic differences within South Florida, this will not differ significantly. Thus, comparing the isotopic range from several populations can only be environmentally induced.

> As stated before discussion on bulked valve measurements will be considered.

Page 16 Lines 11-12. I could not see evidence in the cited paper for a 1‰ range in lake water in Lake Okeechobee. For sure, there are a few datapoints from this lake in Figure 4 of Harvey and McCormick, but it is unclear whether these are representative of seasonal variations in the isotopic composition of Lake Okeechobee or are samples from different locations in the lake from one single sampling period. Also, by themselves, these data do not support the suggestion that the amount of evaporation of this lake is seasonally constant (see also my point above).

> Considering the uncertainty of the referred data, this sentence will be rewritten.

Page 17. Section 5.3. This method will only work if you can assume that the isotopic composition of water at each of your sites is closely coupled to the isotopic composition of rainfall. As noted above, you have very little evidence to support this. As I also note above, figure 7 is the key figure in the paper and it suggests to me that the specimens of Cytheridella ilosvayi may have calcified quite close to the sampling time. In this case, I wonder whether these theoretical calcite calculations are really very necessary.

> As I noted above, rivers are influenced by the isotopic composition of surface runoff from precipitation and groundwater inflow, which is again seasonally controlled by precipitation.

> Respective references and comments will be given to support the applied model approach and qualitative discussion on the influence of evaporation will be given.

Page 18. Lines 17-19. I am not convinced that this is valid, because your calculations of theoretic equilibrium calcite composition assume that you know the isotopic composition of the water at each site in detail, and I am not convinced that you do.

> As every model approach, the calculation is an approximation of the real situation at any site. Any other comparison of theoretical calcite from environmental studies has to deal with assumptions to overcome uncertainties between the calcification time and sampling (e.g., 'best fit' correlation of Decrouy 2009). As shown before, the isotopic variation of rivers mainly depends on the isotopic composition of groundwater and precipitation. Precipitation is also the main source of groundwater and its isotopic composition develops similar to precipitation will lower amplitudes. We assume that the monthly variation of river water is restricted to the isotopic variation of precipitation. Then, our calculations represent the highest possible variation of the theoretical calcite. However, at page 18, lines 20 to 30 we discuss the reduction of the variation of the calculated calcite (by a high influence of groundwater) and the possibility of other calcification times for Cytheridella during the year.

Line 34. Cytheridella ilosvayi may be a neotropical species, but Florida is a subtropical locality and the seasonal variations in temperature, although certainly not as large as at higher latitudes, are not negligible. In any case, it is misleading to state that your work shows that 'tropical' ostracods show seasonality.

> As stated before, the southern part of Florida is tropical indeed and temperature differences will be discussed in more detail.

General comment. You emphasize oxygen-isotopes in your paper. Although the carbon-isotope results are referred to, your comments about these are more limited and speculative, not least because you have very little information about the variations in the carbon-isotope composition of the DIC. It would be useful to comment on the carbon-isotope data in a little more detail.

> More detailed discussion will be given.

Technical comments

Page 2 Line 3 and elsewhere. von Grafenstein (not Von)

> This will be changed.

Line 10. No need to include the word 'stable'

> This will be deleted.

Page 5 Lines 2-4. What has happened here?

> This was a relic from the manuscript preparation from the Copernicus Word template and will be deleted
* * *

---

## Referee Comment (RC2) · Anonymous Referee #2 · 2 May 2017

The manuscript by Meyer et al. entitled "Also tropical freshwater ostracodes show a seasonal life cycle" presents important information on (sub) tropical ostracode calcification and related environmental information. Presented data and results are novel and will serve in future (sub) tropical paleoenvironmental reconstructions. However, there are important details that need to be improved before. I have carefully read the comments from the other reviewer and I agree with his comments and concerns that authors should carefully and almost completely consider. I therefore, tried to focus on other aspects in order to not be repetitive. I attach a pdf file with my comments and corrections and hope they will help the improvement of this manuscript before its future publication.

Please also note the supplement to this comment:

[Figure]

http://www.biogeosciences-discuss.net/bg-2017-38/bg-2017-38-RC2-supplement.pdf

**Supplement:**

[revised manuscript text omitted]

---

## Author Comment (AC1) · 5 Jun 2017

General Comments from the Authors

We thank the anonymous referee for the critical and helpful comments on our open discussion paper. Many comments are similar to the once of the first referee J. Holmes.

The major concerns of the referee refer to the inexact formulation of the aims of the study and related discrepancies in the whole manuscript due to imprecise and/or missing information in the abstract, introduction and methods that are relevant for the structure of the paper and further interpretations in the manuscript. Many comments suggest, that the separation between the spatial comparison of the investigated sites based on our data, and the modelling of calcification periods based on instrumental

background data is unclear. Thus, some of the referee comments arise from the mis-understanding which data have been used for the single parts of the study. Minor concerns have been given to the differentiation between juvenile and adult isotopic measurements, the averaging problem of sample bulking and some structural weaknesses on the seasonal variation of climatic and isotopic background data and parts of the discussion.

We checked the manuscript carefully and will edit it respectively to the suggestions of both reviewers. This includes the following points: The title will be changed to "Modelling calcification periods of Cytheridella ilosvayi from Florida based on isotopic signatures and hydrologic data" to point out the most important discovery of this study. Further, the abstract and introduction will be rewritten to add missing information suggested by the referees and reformulate the aims of the study to differentiate clearly between the spatial comparison of ostracod shells and their host waters and the subsequent seasonal modelling for river samples based on the within-sample variability and instrumental background data. Additionally, further information on the model will be given to better explain the significance of the model. In addition, discussion will be added referring to the difference between juvenile and adult isotopic measurement and the effect of "bulking" on the isotopic variation. Based on this discussion samples CAL-1 and CAL-2 are unsuitable for the modelling of calcification periods and will be excluded. Thus, Figure 8 has to be changed. According to the suggestions of the referee and the changes mentioned above, the conclusions have to be rewritten. Finally, spelling mistakes will be corrected according to referees and beyond.

More detailed comments will be given point by point as received in the referee comments in the following pdf file

Please also note the supplement to this comment:
http://www.biogeosciences-discuss.net/bg-2017-38/bg-2017-38-AC1-supplement.pdf

[Figure]

**Supplement:**

**Reply to anonymous Referee Comment**

Title

tropical/sub-tropical?

> ➤ The title will be changed. Detailed information on the climatic conditions of Florida will be given in section 2 of the manuscript.

I am not 100% convinced about this title, there is some info on tropical ostracodes that development is related to seasonal changes, so I do not see how this title/manuscript would show novel results...I could give a list of some papers but authors should as well do a better research on this. env variables related to seasonal changes are temperature, mixing of lakes, precipitation/begin of rain, please search for this.

> ➤ We did extensive research on life cycles and calcification of tropical freshwater ostracods and environmental parameters influencing them, without success. A second research including the headwords suggested by the referee was without success. It would have been helpful, if the referee would have given at least one or two concrete references on the topic.

I would really try to highlight the most important discovery of this manuscript in order to provide an improved title.

> ➤ The title will be changed.

Abstract

didn't see anything regarding d13C isotope results, as well as some background info as for d18O, please add. would be important to mention why did you run analysis on C. ilosvayi and not other species

> ➤ Additional information recommended from both referees will be added in the Abstract

Line 8 and larval stage? other factors? vital effect? reproduction, food sources, etc.?

> ➤ The abstract will be rewritten and this particular sentence will be deleted. However, larval stages are included in the term species. Vital effects are species specific and considered to be constant within one population and do not influence the maximum variation. The influence of vital effects is included in the discussion. Food sources are not relevant for the oxygen isotope composition of the shell. For carbon isotopes the dissolved inorganic carbon in the water is the main source for the formation of the shell. The influence of the ostracod diet is unclear and might be very complex. Missing background data don't allow a proper discussion on that. We are not sure what is meant with reproduction. If calcification time is meant, still the sum of environmental changes will define the isotopic variation of a population and this is restricted to the time in which the population calcifies.

Line 11 here you need to say when you took samples, since the focus of the paper is seasonal, you need to say when samples were taken, where, and months? years? because you mention something on seasonal changes at the end of the abstract, which is already too late.

➢ The seasonal approach is based on a model with different assumptions on the general seasonal development of rivers and their relation to ostracod isotopic signatures. The seasonal data are obtained from the literature and independent from our sampling. Thus, timing of sampling is not the key to the seasonality in this case. A clear separation of this will be given in the abstract.

what kind? only rivers? lagoons? lakes? rivers? try to give the reader as much info as possible

➢ This information will be added to the abstract.

what kind of instrumental data? physical? chemical? both? be more specific

➢ This information will be added to the abstract.

a wider region or as well south florida?

➢ The same region was meant, this sentence will be rephrased.

Line 17 only rivers were studied?

➢ In the first part of the study all sites are compared, while for the model only rivers are used. Respective sentence will be rephrased to make that clear.

Line 21 need to explain before when you took samples and that sampling was seasonal

➢ As stated above, the seasonal calcification period is based on the model that was applied using literature data on the seasonality.

Line 23 how? how does your study improve future studies, paleontology is too general, maybe it would make the manuscript stronger if you could give precise use of such studies, rather than mention a too general use.
related seasonal

➢ More precise use of this study for paleontology will be given in the manuscript. The investigation of the within sample-variability of modern ostracods in relation to environmental changes is the basis for the interpretation of high frequency climate variabilities in the ostracod record. Such paleontological studies have already been performed by Escobar et al. (2010) and Dixit et al. (2015).
➢ However, this sentence will be deleted

Line 7 describe

➢ This sentence will be rephrased. Vital effects are discussed in section 5

Line 16 until now I do not know how often did authors collect samples, it seems that the temporal scale is so important but it is needed to describe this in the abstract

➢ The separation between the seasonal data for the model and our data will be described more clearly in the abstract.

Line 22 and d13C?

➢ The model only refers to oxygen isotopes.

but in the abstract you mention temperature, that is not chemical

➢ The temperature mentioned in the abstract refers to seasonal data revealed from the USGS climate viewer, not to our own measurements. However, as we also did temperature measurements this will be changed to physico-chemical in the abstract.

Line 25 years? same temporal scale/simultaneously to sample collection?

➢ Exact information on the seasonal data origin is given in the Methods and Table 3 and 4.

it is also important to mention in the abstract how you collected ostracodes (surface samples or hand net, since you say that they are "living ostracodes" then this is highly relevant. as well did you analyze only adults or each larval stage? or mixed?

➢ For the abstract the description "living ostracods" is in our opinion appropriate, more details on sampling are given in the methods.
➢ We analyzed mainly adults, some juveniles are also included, information will be given in the abstract and discussed later

Line 4 in the abstract one can understand that you only studied rivers therefore include more info there

➢ As stated above, the separation between the two parts of the study will be given in the abstract

Line 5 are you sure that one can talk about seasonality only with two samplings?

➢ This comment probably also results from the unclear separation between the regional comparison of samples and the model approach which is based on monthly data from the literature.

a more robust study would include 2 years or 1 but more frequently sampled...justify it please. Why those months? are they really the two months in a year that are more significant to prove your hypothesis? I would as well try to include a sentence in the abstract what is the scientific question you are trying to answer, so that it does not sound as a methodological manuscript.

➢ Repeated sampling would be desirable, but, especially in regional studies it is too time consuming and expensive and in many cases not realizable. This is exactly the point of the study. When no temporal/seasonal information of the ostracod species is available, than the isotopic composition

and range of an ostracod population can reveal information on the calcification time and duration. The seasonality is not based on the two samplings, but on the model that was applied to calculate theoretical calcites and the comparison the isotopic ranges of C. ilosvayi.

Line 31 Since you collected samples from a wide variety of aquatic ecosystems, displaying different chemical properties and probably physical, makes it more difficult to understand which env. parameter is/are resposible for ostracode seasonal changes...please discuss this later

➢ A paragraph will be added in section 5.1

Line 5 I would clearly indicate which months are considered winter, summer, spring, fall and dry and wet season...all is important for your results and discussion, part of the info is here but not in a clear way

➢ This section will be restructured to point out important influences on the seasonality of surface waters

then why didn't you take samples in these months? just try to explain why do you believe that the months you sampled, should be the right ones, what was the criteria or how can you prove that your methods are correct/reliable.

➢ When no information on the life cycle of a species is available then the right time for sampling is uncertain. Sampling in the warmest or coldest month does not necessarily mean to be more successful. But, as stated before, the timing of sampling is not the important point for the study, but the within sample variation that reflects the conditions during calcification time of C. ilosvayi

Line 29 info on d13C, sometimes d13C show a more clear seasonal effect

➢ In this section we give information on the seasonal variation of isotopes in Florida obtained from the literature. As stated in the last paragraph of the section, data on the d13C composition in Florida are rare. But, general information on seasonal variation of different environments could be added.

Line 24 I know that this is a benthic species, but using a ekman grab? did you use one? please describe which equipment you used so that authors do no need to go to Meyer et al. 2016 to see this. Living ostracodes or more reliable samples are those from hand nets, because generally when you use an ekman grab even if you say that one uses the top cm, those are always some years or even decades. Since this is not a paleoenvironmental study, it is not possible for you to know sedimentation rates, but if you are talking about living ostracodes, then the way to do it would have been to collect ostracodes either way, and directly see them under the stereoscope if they were actually living. If not, then I would trust hand net samples more than ekman samples. Please add information and justify methods used.

➢ We used a hand net scratching over the surface to get sediment material from the upper ~2cm and to sample living animal.

> To refer to former studies to get more detailed information on the methods is a common procedure, however, relevant information on sampling are added

adult, juveniles? all? male, female?

> We found female, male and juvenile stages. Information on that will be added in the methods and supplementary material and discussed in section 5.

Line 25 shortly mention in abstract

> Will be added.

Line 27 how?
these samples need to be fixed, what did you use? otherwise isotopic signatures are changed by biological activity

> This is described in Spötl (2005), which we also sited: "…Prior to going into the field, the exetainer is preloaded with five droplets of phosphoric acid (ca. 90%), capped and the headspace is flushed with He 6.0 in the autosampler by penetrating the butyl rubber septa of the exetainers' disposable caps…"
> Just in case the referee refers to the ostracod material and not to water samples: surface sediment samples containing ostracod material were stored in ethanol (90%) to preserve soft part material until picking. A respective sentence on that will be added

Line 5 until now it was clear to me that d18O was measured, d13C as well but lots of information is missing, and dD is new to me. Did you as well determine this and are you using it for your discussion? as well as DIC? then you need to include this info in the abstract

> the correlation of dD and d18O was used to show the GMWL and the deviation of the water isotopes along a local evaporation line. dD will be added to the abstract.

Line 4 is there more biological information for this species in the literature? if so then add in the introduction. maybe since this is a widely distributed species in the entire neotropical region.

> There are some information on the biology and geographical distribution available and will be added in the manuscript.

I think that in the introduction the authors needs to make a better job selling the topic to non-experts, or making it more interesting to a wider interest, more info on ostracodes, why this species and not others, why is this study new? what is different from other past studies?

> The introduction will be rewritten considering the critical and helpful points of the referee.

Line 8 how long does this species live? some info about that? what does the literature say?

> To our knowledge, there is no information available on the life span of C. ilosvayi. There is some information on seasonal abundances from different localities from Purper (1974), Higuti et al. (2007), Pérez et al. (2011). These information will be added in the introduction.

is this calculation meaningful? what if calcification is longer? and different per type of aquatic ecosystem,

> Our model approach has not been tested before. Climatic data are commonly given in monthly time intervals, and a variation in monthly precipitation and temperature data can be seen in Florida, so this seemed to be a realistic starting point for calculations.
> A longer duration of the calcification is already considered in the discussion.

you have until now, not said if this applies only for adults (what makes more sense) or all larval stages. How many samples/ostracodes did you use per analysis? I hope single valves, since the focus of this paper is isotopic...if you mix 2 or more then you have a slight error, and if different samples/analysis have different number of valves then it is hard to compare among analysis right?

> Most of our measurements were performed on adults, but some also on the A-1 juvenile stage. The latter once are excluded from the model. The number of measurements and number of valves differed between samples, depending on the material that was provided. The number of measurements is given in the supplementary material.
> The averaging caused by bulking will be added in the discussion.

Line 26 why a long period? and not the years where you took samples? what is the idea behind this? explain please.

> Measurements of precipitation are not distributed equally (see Table 4). We included all available data to get significant monthly values

Line 4 did not mention this before, remember that all needs to be explained in methods and no surprises when you are in results. only present in results, things that you have mentioned before, no new things.

> The TDS is calculated from the ion concentrations, this will be added in the methods

water?
> Yes, this should be already clear from the methods.

Line 30 I would use same number of decimals for isotopes and rest of environmental information...1 decimal is probably fine and please always add a minus or plus before

> According to the referee, numbers of environmental parameters are adapted to one decimal for all isotopic data and most chemical and physical data.
> Minus and plus marks will be placed in front of isotopic values, while isotopic ranges will not be marked with any sign

Line 15 add plus throughout the manuscript

➢ Will be added.

Line 8 I would first discuss your data and then complement with other references or give the interpretation/explanation supported by references later

   ➢ The section will be restructured according to the referee

Line 5 total?
   ➢ Yes, this is the total amount of phosphate. Respective sentence will be changed.

Line 6 are these your own data? if not then cite

   ➢ Yes, they are. This will be indicated.

Line 6 more sources in Leng et al.

   ➢ The referee probably suggests the addition of Leng and Marshall (2004), which will be added. If another reference is suggested, the year of the reference would have been helpful.

Line 7 use same nr of decimals throughout the manuscript

   ➢ Will be changed.

Line 31 showed

   ➢ Will be changed.

Line 17 I would like that authors elaborate more this idea, because I do believe that it is highly important to consider larval stages and parallel run lab studies of species life cycle for a robust interpretation

   ➢ The idea will be extended in the introduction, also some more information on that are already given in section 5.2.1.
   ➢ Laboratory studies are hardly comparable to natural environments and are mostly designed to test changes of single factors. In addition, lab studies are not always possible to perform, as in our case, e.g., because transport of living individuals to our lab was not possible. But, our approach is based on single sampling, and juvenile material was limited to a few individuals and not enough to run isotopic measurements. Besides, there are still uncertainties concerning biomineralization of juvenile stages which may influence isotopic results.

Line 21 sentences should not start with abbreviations in such cases full species name should be written

➢ This will be changed.

Line 11 Pérez et al. (year)

➢ This will be changed.

Line 14 as well adults? female? I have not seen much information on larval stages in your study, which would have been a tremendous contribution

➢ information on juveniles are scattered, but will be added.

Line 21 why only rivers? what about the rest of aq. waterbodies?

➢ Rivers are mainly influenced by temperature and precipitation, while in other waterbodies evaporation is an additional important factor. As we have information on the d18O composition of rain, calculation of theoretical calcite in rivers is possible

Line 34 not true...there is some information, however these topic is not the focus of the papers so one need to carefully read the scientific papers in order to get information on life cycles.

there is info on conc. dissolved oxygen, temp, rains, maybe some in spanish but one cannot generaliza that there is no published information. Ask directly the experts from middle and southamerica and you will find out more.

➢ As stated above, we did extensive literature research (also with the suggested headwords) without success. Some specific literature suggestions from the referee would have been very helpful.

Line 10 write full name

➢ This will be changed.

Line 18 I find this part too long. Most information could go in abstract, since are the main results, and here authors should rather present 3-4 main conclusions, what did they discover that will benefit the community working with this discipline? what is new from other studies? this part should not be similar to the abstract

➢ Conclusions will be rewritten and some information will be included in the abstract.

Line 21 why is this important? how will this improve paleostudies?

> When calcification of a species is seasonally restricted the isotopic signatures that are reflected by that species will differ from a species which calcifies during another season or during the whole year. This information can be used for a better interpretation of high-frequency climate changes in paleontological studies (e.g., Escobar et al. 2010, Dixit et al. 2015)

Line 31 could be? or were? remember that these are conclusions...

> This will be changed to "were".

---

## Author Response (AR1)

**Reply to Reviewer Comments**

**General Remarks from authors**

We received two reviews for the first version of this manuscript, one from Jonathan Holmes and another one from an anonymous reviewer. We followed the suggestions of the reviewers and revised the manuscript carefully. Most of the reviewer suggestions were adopted as requested and further modification was carried out to make the manuscript more coherent for reading.

Preliminary comments on reviewer remarks were already given point by point as authors' comments during the open discussion process. The final changes that were made can differ from the authors' comments in particular points (e.g., additional information are not given as commented in the discussion, but were added in the methods or results).

Here we provide an overview about the most important changes that were made.

Enclosed to the following reply we send the revised manuscript with tracked modifications.

**The following changes were made:**

Many parts of the manuscript were restructured and therefore paragraphs and sentences are added, changed, shifted or deleted throughout the whole manuscript to make it more coherent.

**Title**

The title of the manuscript was changed to highlight the most important outcome of this study as suggested by both reviewers.

**Abstract**

The abstract was adapted according to changes that were made in the manuscript. More details on the included data were given as suggested by both reviewers.

**Introduction**

The whole introduction was rewritten to better point out the aims of the study. This includes the idea of single shell measurements as indication for a seasonal restriction of molting periods and many ecological details on *C. ilosvayi* as requested by the anonymous reviewer.

**Study area**

The whole chapter was restructured to support assumptions for the presented model approach in this study.

**Climate**

Here we point out the position of South Florida at the boundary of a tropical and subtropical climate and give specific information on the climatic conditions of region and more details on the conditions during the sampling period 2013/2014.

**Hydrology and Study sites**

These sections were combined to one with emphasis to relevant processes on the seasonal environmental variation of the investigated sites.

Information on the deeper aquifer systems was abridged as this is only of limited importance.

10 Some of the details on the study sites were added to table 1. These and other details that are not relevant for the further discussion were deleted from that section.

**Seasonal variation of water isotopes**

This section was restructured for a coherent presentation of available data and to emphasize important information

15 on surface water variation to prove the plausibility of the subsequent model.

**Material and Methods**

Details on sampling and ostracod material used for isotopic analyses were given in the Methods and Results and supplementary material (adult, juvenile, number of valves per measurements).

20 The required number of isotopic measurements in one sample was tested to be representative for environmental variation to exclude 'bulking of shells' as reason to cause reduced variability.

The assumptions for the calculation of calcification periods were clearly formulated

**Results**

25 **Seasonal variation of theoretical equilibrium calcite for river habitats**

The section was separated into smaller sections

**Isotopic signatures of ostracod calcite**

Information on the isotopic composition of juvenile measurements were added and the outcome of the required

30 number of isotopic measurements were added.

**Discussion**

**Physico-chemical and isotopic characteristics of surface water habitats**

The whole section was rewritten to signify environmental parameters influencing the isotopic composition of habitats.

Additional information of potential influences on $\delta^{13}C_{DIC}$ is given.

**Stable isotope compositions of *C. ilosvayi***

Some paragraphs were shifted

**Conclusion**

Conclusions were rewritten to answer the formulated research questions in the introduction.

**Minor points**

Spelling mistakes and other small mistakes were corrected within the whole manuscript.

Number of decimals was adapted to one decimal place. Minus and plus marks were placed in front of isotopic values, while isotopic ranges were not be marked with any sign.
Details on the number of valves and the development stage and sex are added to the supplementary material.

A further supplementary file was added with results of isotopic signatures from further water samples of Pease River and sites within this river basin, Loxahatchee River and Caloosahatchee River.

**Tables**

Table 1: details on sample sites were added.

In all tables the number of decimals was adapted to one decimal place.

**Figures**

Figure 2 was changed to give separated information on the climate 2013 and 2014; the respective figure caption was corrected

Figure 1 (Study area) and 2 (Climate) were swapped

In Figure 7 one missing data point was added

In Figure 8 the sample CAL-1 was excluded from all charts

[revised manuscript text omitted]

---

## Author Response (AR2)

**Authors' response**

We thank Jonathan Holmes for the repeated review of our manuscript. We followed his suggestions revised the manuscript carefully. Here we provide the list of changes given point by point.

line 7: temperature and composition?

➢ Changed as suggested

line 18: no comma after both

➢ Changed as suggested

At the start of the introduction: perhaps add something like 'Ostracodfs are small aquatic crustaceans, which produce shells composed of low-Mg calcite'

➢ Changed as suggested

line 18: von

➢ Changed as suggested

line 14: there exists

➢ Changed as suggested

line 15: is it realistic to quote the vital offset to such a high level of precision? Perhaps give the value as about +1 ‰

➢ Changed as suggested in the whole manuscript (page 8 and 18)

lines 10-18: seems strange to mix mg/l and psi

➢ The given units in this section are separated clearly. Salinity values are given in PSU while all other hydrochemical parameters (TDS, Ions) are given in mg/l (see also table 2). In our opinion there is no need to change the units, as they are common for these parameters.

line 13: Omax and Omin look odd- use d18Omax and d18Omin (with delta signs and superscripts of course!)

➢ Changed as suggested.
➢ Fig. 3 had to be changed to include the changed notation.

line 25: perhaps range is a better word than variation

➢ Changed as suggested

line 30: it is about 0.2‰ - the relationship is not perfectly linear

➢ Changed as suggested

line 22-23: this statement needed to come earlier in the MS

➢ The sentence was shifted to chapter 3.4 Calculation of calcification periods (c)

line 9: lower ionic concentration?

➢ Changed as suggested

line 14: temporally restricted

➢ Changed as suggested

line 25: this sentence doesn't quite make sense

➢ The sentence was rephrased.

line 10-14: I do not believe you can with any confidence that these differences provide any indication of the vital effect because you have very limited constraint on the water temperature and isotope composition at the time of shell formation.

➢ Positive vital effects are reported for several species.  In the case of C. ilosvayi, a vital effect of about 1 ‰ is known (Escobar et al., 2012). The existence of this offset during the shell calcification and its constancy is one of the assumptions for this modelling approach (see chapter 3.4). Thus, a positive deviation of the monthly calculated calcite and the mean isotopic values of C. ilosvayi have to be considered for a plausible calcification time.
➢ However, the perspective sentences have been rephrased to make that more clear.

Also, do the d13C values help with you working out the timing of calcification? Even if they do not, it would be worth stating this.

➢ Unfortunately, for this study it was not possible to confine the timing of calcification with d13C values. However, a sentence on the potential use of d13C was added.

line 21: rainfall events

➢ Changed as suggested

line 26-27: consider rephrasing

➢ The sentence was rephrased as suggested

**Further changes**

Figure caption

Fig. 3 Spelling mistake was corrected